# Reanalyzing the spatial representativeness of snow depth at automated monitoring stations using airborne Lidar data

Jordan N. Herbert[1], Mark S. Raleigh[2], Eric E. Small[1]

[1]Department of Geological Sciences, University of Colorado Boulder, Boulder, 80309, USA
5  [2]College of Earth, Ocean, and Atmospheric Sciences, Oregon State University, Corvallis, 97331, USA

*Correspondence to*: Jordan N. Herbert (jordan.herbert@colorado.edu)

**Abstract.** Automated snow station networks provide critical hydrologic data. Whether point observations represent snowpack at larger areas is an enduring question. Leveraging the recent proliferation of airborne Lidar snow depth data, we revisit the question of snow station representativeness at multiple scales surrounding 111 stations in Colorado and California (U.S.A.) from 2021–2023 (n= 476 total samples). In about 50% of cases, station depths were at least 10 cm higher than areal-mean snow depth (from Lidar) at 0.5 to 4 km scales. The nearest 50 m Lidar pixels had lower bias and were more often representative of the areal-mean snow depth than coincident stations. The closest 3 m Lidar pixel often agreed with station snow depth to within 10 cm, suggesting differences between station snow depth and the nearest 50 m Lidar pixel result from highly localized conditions, not the measurement method. Representativeness decreased as scale increased up to ~6 km, mainly explained by the elevation of a site relative to the larger area. Relative values of vegetation and southness did not have significant impacts on site representativeness. The sign of bias at individual snow stations is temporally consistent, suggesting the relationship between station depth and that of the surrounding area may be predictable. Improving understanding of snow station representativeness could allow for more accurate validation of modelled and remotely sensed data.

## 1 Introduction

Mountain snowpack provides water to over a billion people worldwide (Dozier et al., 2016) and comprises approximately half of freshwater available in the western United States (Li et al., 2017). Snowmelt impacts agricultural activity (Qin et al., 2020), ecosystems (Blankinship et al., 2014; Dollery et al., 2006), and influences the magnitude and frequency of natural hazards such as wildfires, floods, and droughts (Dierauer et al., 2019; Musselman et al., 2018; Westerling et al., 2006). The amount and timing of water availability in snowmelt dominated watersheds is dependent on snowpack characteristics. Despite recent advances, existing remote sensing techniques do not allow for spatially and temporally continuous monitoring of snow water equivalent (SWE) in the complex terrain of mountain watersheds (Lettenmaier et al., 2015). Instead, assessments of water stored in mountain snowpack for hydrologic research and applications (e.g., streamflow forecasting) rely on a combination of ground-based snow sampling, remote sensing, and modeling (Pagano et al., 2009).

Automated stations (hereafter: snow stations), such as the Natural Resource Conservation Service's (NRCS) Snow-Telemetry (Snotel) network, provide temporally continuous, high-quality measurements of snow depth and SWE at over 900 locations throughout the western United States. Snow stations are strategically located to maximize their utility for water supply forecasts. Sites with more persistent snow (e.g., higher elevation, northern aspects) are preferred, since locations with more persistent snow provide data for streamflow forecasts longer into the ablation season (NRCS, 2011). Stations are built on flat surfaces, below tree line (between 2745–3350 m above sea level), and in areas shielded from high winds (Molotch & Bales, 2006; NRCS, 2011; Woelders et al., 2020). The specific requirements for snow station locations, combined with their uneven distribution across the landscape, may increase the potential for bias when using station data to represent larger areas such as an entire watershed.


In addition to aiding water supply forecasts, snow station data have been applied to a wide array of applications in snow hydrology. Snow station data are frequently used to validate models (Pan et al., 2003; Schneider and Molotch, 2016) and as ground truth references for remotely sensed data (Klein and Barnett, 2003; Lievens et al., 2022; Painter et al., 2016). In these cases, station data are used as the "true" values against which the model and remotely sensed data are validated. However, the

datasets being validated frequently represent areas on the hundred-meter to kilometer scale, much larger than the ~1-3 m sampling area of a snow station. Another common use for snow station data is input for data assimilation frameworks (Dechant and Moradkhani, 2011; Margulis et al., 2019; Slater and Clark, 2006; Smyth et al., 2020; Barrett, 2003). These applications also apply snow station data to represent the (usually much larger) scale of the model resolution. Finally, station data have been spatially interpolated into gridded products (Broxton et al., 2019; Molotch et al., 2005; Lopez-Moreno et al., 2010). Even

though the interpolation may include the influence of landscape factors such as elevation or aspect, the representativeness of the snow station data is typically unknown and is thus unaccounted for in the interpolation scheme.

Care is warranted when extrapolating snow station data to larger areas because the distribution of snow across a landscape can be highly variable, especially at meter-to-hundred-meter scales (Blöschl, 1999; Clark et al., 2011; Scipión et al., 2013). As a

result, many studies have assessed the utility of point data to represent larger areas. Evaluations of point measurement representativeness suggest single measurements are inadequate to represent areas as small as 10 $m^2$ (López-Moreno et al., 2011) or 30 $m^2$ (Fassnacht et al., 2018), and over 50 point measurements are required to represent an area of 300 $m^2$ (Watson et al., 2006). Other investigations used manual sampling of snow depth and SWE combined with binary regression trees to determine how snow properties vary surrounding a limited number of snow stations (Meromy et al., 2013; Molotch & Bales,

2005). These results suggested that half or fewer of stations yielded snow depths within 10% of the mean snow depth of the surrounding area (areal-mean snow depth). Embedded sensor networks surrounding an operational snow course and snow station demonstrated that neither the snow course nor the station represented the areal-mean snow depth to within 20-30% at the 1 $km^2$, 4 $km^2$, or 16 $km^2$ scales due to differences in the surrounding topography (Rice and Bales, 2010).

Other studies have used high spatial resolution mapping of snow depth from airborne Lidar to assess snow station representativeness, though these efforts were limited in scope. Grünewald and Lehning, (2011) used data from five snow stations and three Lidar surveys to assess if snow stations can accurately represent the change of snow depth with altitude. Grünewald and Lehning, (2015) used Lidar surveys from six different watersheds (one survey per watershed), finding sites that met the criteria for snow station locations (as opposed to using real station data) to assess snow station representativeness.

These efforts found that snow stations typically overestimate SWE, possibly due to the sampling locations occurring on flat terrain compared to the more characteristically sloping mountainous terrain of the surrounding area. Of the sites that were

deemed representative of the surrounding area (within 10% of the areal-mean), there were no discernible similarities in topographic attributes that would serve as a predictor for "well-placed" sites.

The aforementioned studies were limited in the quantity and spatial-extent of study areas due to the labor requirements of manually collecting samples and the limited availability of high-resolution Lidar snow depth data. The recent proliferation of Lidar snow depth data in the western U.S. made possible by the Airborne Snow Observatory (ASO; Painter et al., 2016) provides an opportunity to assess the representativeness of snow monitoring stations using high confidence, spatially distributed Lidar data that are collocated with snow station locations. We utilize Lidar snow depth data available in watersheds

in Colorado and California to revisit the question of how representative the locations of snow monitoring stations are compared to the surrounding area and whether the relationship is consistent over time.

Here, we address the following questions: (1) How variable is Lidar snow depth around operational snow stations? (2) What is the distribution of relative snow depth values (RSD; defined in Section 2.2.2), and how does RSD change when calculated

for different spatial scales and point snow depths derived from different sensing techniques (i.e., in-situ versus remotely sensed)? (3) Do individual sites demonstrate repeatable patterns of RSD sign and magnitude over time? (4) What impact do relative landcover and topography variables (specifically, elevation, fractional vegetation, and southness) have on RSD? While answering these questions we focus on snow depth, not SWE, because snow depth is the variable measured directly both by airborne Lidar and at snow stations. See Section 2.1.2 for further explanation on this decision.

**2 Methods**

**2.1 Study sites and data**

We selected locations in Colorado and California that have coincident airborne Lidar and snow station data over the interval February 2021 through June 2023. In Colorado, we utilized 40 Lidar surveys in 13 watersheds, containing 48 active snow stations, totaling 138 instances of coincident Lidar and snow station data. All Colorado Lidar surveys were carried out in April

and May, typically with two surveys per year per basin. More data were available in California, where we utilized 108 Lidar surveys in 13 watersheds, containing 63 active snow stations, totaling 338 coincident Lidar-station comparisons. California surveys were conducted between January and June, with most surveys between March and May. Locations of the Lidar surveys and snow stations are summarized in Fig. 1 (and supplemental Tables 1 and 2). Between both states, we analyzed 476 instances of coincident Lidar-station data.

In the remainder of Section 2.1 we provide detailed descriptions of the datasets we employ in this investigation and the scales at which we employ them.

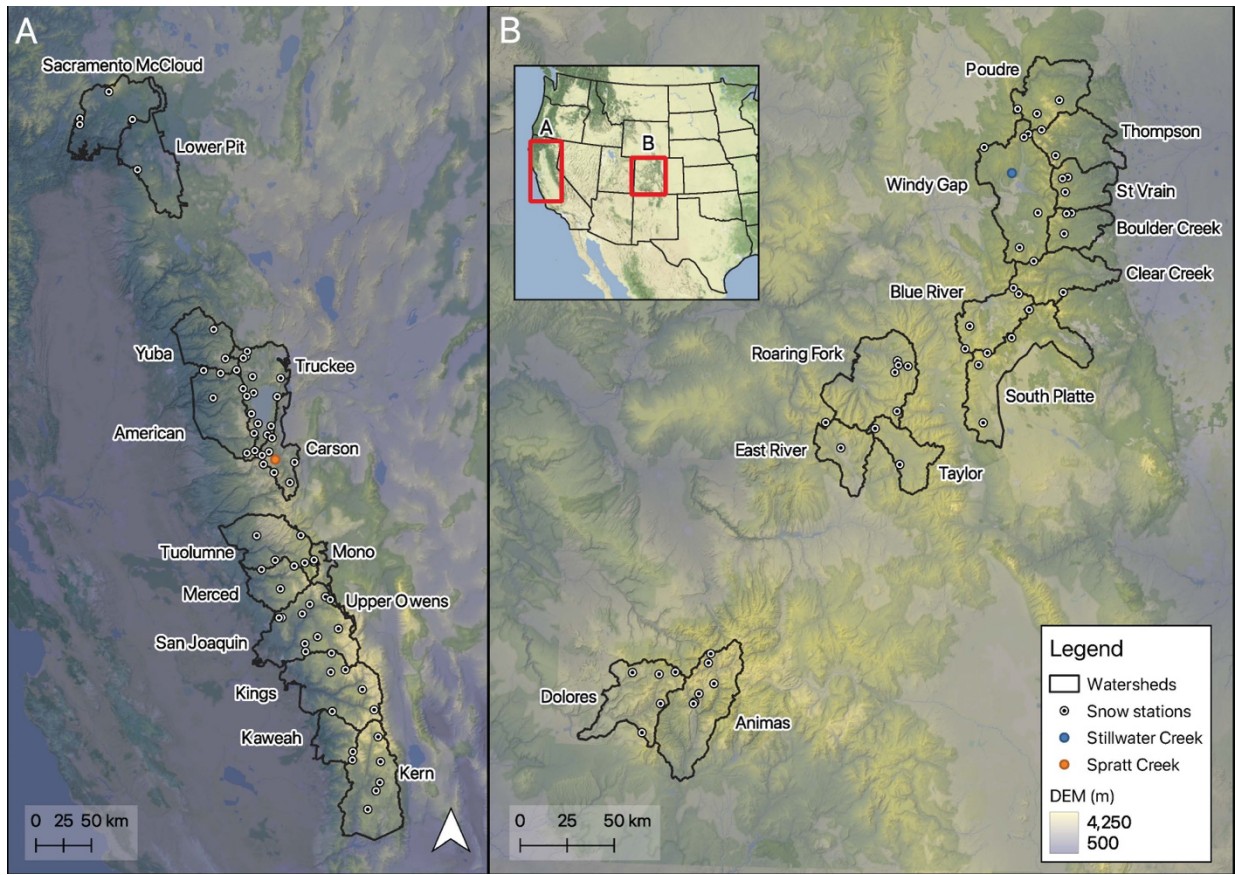

**Figure 1: Locations of Lidar surveys and snow stations in (a) California and (b) Colorado, with watersheds labeled.** Two stations (Stillwater Creek and Spratt Creek) are highlighted as these are used in subsequent examples.

### 2.1.1 Snow station data

The NRCS and the California Department of Water Resources (CA-DWR) operate snow stations which monitor snow depth, SWE, and meteorological parameters at select locations in snow-dominated watersheds. These stations collect snow depth data using an ultrasonic sensor (precision: 13 mm) and SWE data by measuring the mass above a snow pillow (precision: 2.5 mm) (NRCS, 2011). Sensor precision values are not reported by CA-DWR, but should be similar to the NRCS values since they use similar equipment. The typical spatial support (Blöschl, 1999) is 9 m$^2$ for SWE (snow pillow) and ~1 m$^2$ for depth (ultrasonic sensor).

Although SWE is the critical variable for understanding water storage, we conduct our analyses using snow depth because it is the variable directly retrieved by Lidar surveys. Lidar SWE products use modeled density (Painter et al., 2016), increasing

the uncertainty of the measurement as compared to snow depth. Of the existing literature, one study (Molotch and Bales, 2005) directly measured SWE using a federal sampler to get distributed measurements of SWE, but was limited by the total amount of samples collected. Most other studies (e.g., Grünewald and Lehning, 2011, 2015; Meromy et al., 2013) converted snow depth to SWE by assuming a uniform snow density across the study site. Snow density is not uniform across the landscape

and may contribute considerable uncertainty in SWE estimations based on Lidar data (Meehan et al., 2023; Raleigh & Small, 2017; Wetlaufer et al., 2016). Converting values to SWE by assuming a uniform snow density increases the potential error as compared to retaining the values as snow depth. Thus, we keep our analyses in terms of snow depth. Any results herein would be identical if we converted to SWE by multiplying snow depth with a chosen snow density (e.g., Grünewald and Lehning 2011, 2015; Meromy et al., 2013).


We downloaded daily NRCS Snotel and CA-DWR snow depth data from all sites within the bounds of watershed areas surveyed by ASO with airborne Lidar in Colorado and California from 2021 to 2023. We acquired site coordinates (latitudes and longitudes) from the NRCS and CA-DWR websites. Due to the importance of accurate location data for this study, we verified the locations of each snow station using visual inspection of high resolution satellite imagery in Google Earth. We

updated site coordinates in locations where the provided coordinates were visibly offset from an identifiable snow station. The coordinates were updated to the fifth decimal place in decimal degrees, providing ~1 m accuracy for the location of the center of the snow pillow. We assume that the depth sensor is located over the center of the pillow (which can be identified in the satellite images), although we recognize that this is not always true. The location of four CA-DWR sites within Lidar-surveyed watersheds could not be verified and were excluded from the analysis. Site coordinates are available in Supplemental Tables

1 and 2.

We carried out quality control on the snow depth data to ensure accuracy. NRCS data were free from obvious error, while CA-DWR data frequently displayed unnatural jumps in snow depth. In many cases, the snow depth sensor recorded meter-scale changes in daily snow depth, often followed by a change in the opposite direction of the same magnitude. This likely results

from a lack of quality control measures conducted on CA-DWR snow depth data. We discarded clearly erroneous data that recorded unnatural multidirectional shifts of greater than 0.5 m. Upon visual inspection of the data, the 0.5 m threshold removed the unnatural shifts in snow depth.

### 2.1.2 Lidar data

We utilize all ASO Lidar snow depth data available in Colorado and California from 2021–2023. These datasets are available

as gridded rasters at 50 m and 3 m resolutions in the Universal Transverse Mercator (UTM) coordinate system, WGS84. The 3 m product is produced by taking the difference between snow-on and snow-off point clouds and the 50 m product is an aggregation of the 3 m data (Painter et al., 2016). We use the 50 m datasets to analyze the distribution of snow depth surrounding a snow station and calculate the areal-mean snow depth at a range of larger scales (analyses discussed in Section

2.2). The 50 m scale is sufficient to capture snow depth distribution across the landscape at coarser analysis scales and requires

much less storage and computational expense to manage compared to the 3 m datasets. We employ a subset of 3 m gridded snow depth data, extracting the pixel coincident with the snow station.

Snow depth is retrieved from Lidar data by calculating the difference in surface elevation between snow-on and snow-off surveys. The 3 m snow depths record mean absolute errors of <8 cm, and 50 m snow depths record mean absolute errors of <2

cm (Painter et al., 2016).

It is worth noting that we do not exclude any Lidar data based on proximity to human activities (e.g., compacted snow in ski areas, deeper snow due to snow-making, snow removal on roads) which may impact areal-mean snow depths. Snow stations are often built in secluded locations which we expect are minimally impacted by human activities, but this is limited to only

the small (~30 m) area surrounding a snow station. Lidar surveys encompassing ski areas, towns, and roads have the potential to record snow depths that do not represent the "natural" snow depth that would have been measured in the absence of human impacts. We chose to not remove any Lidar surveys due to the difficulty of finding an objective method to do so, and the changing degree of human impact at a site with scale. We found that at least eight snow stations are near ski areas, but did not find a consistent bias in the snow depths across those sites.

**2.1.3 Landcover and topography data**

We obtained digital elevation models and vegetation datasets surrounding all snow stations employed in this study. For the digital elevation model, we use the 10 m resolution USGS National Elevation Dataset (Gesch et al., 2018). These data are used for their elevation values as well as to calculate southness. Southness serves as a metric for how exposed an area is to solar radiation in the northern hemisphere and is calculated as the sine of the slope multiplied by the cosine of the aspect (Dozier

and Frew, 1990). For vegetation, we downloaded National Land Cover Database percent tree cover dataset (2019), which provides fractional vegetation (FVEG) at 30 m resolution (Dewitz, 2021). We bilinearly resampled all landcover and topography data to match the 50 m spatial resolution of the Lidar data.

**2.1.4 Data representing the snow station**

We use different data sources to represent snow depth at the snow station. In doing so, we can establish if any biases result

from using data with different spatial coverage and sampling methodology. These sources include the reported snow station snow depth (station SD), the coincident 50 m resolution Lidar pixel (50 m SD), and the coincident 3 m resolution Lidar pixel (3 m SD). These data sources have different spatial coverages (1-3 m versus 50 m) and use different sampling methodologies (in-situ versus Lidar). For our analyses we primarily use 50 m SD and station SD; station SD assesses the performance of the station itself while the 50 m SD assesses the general location of the snow station within the landscape.

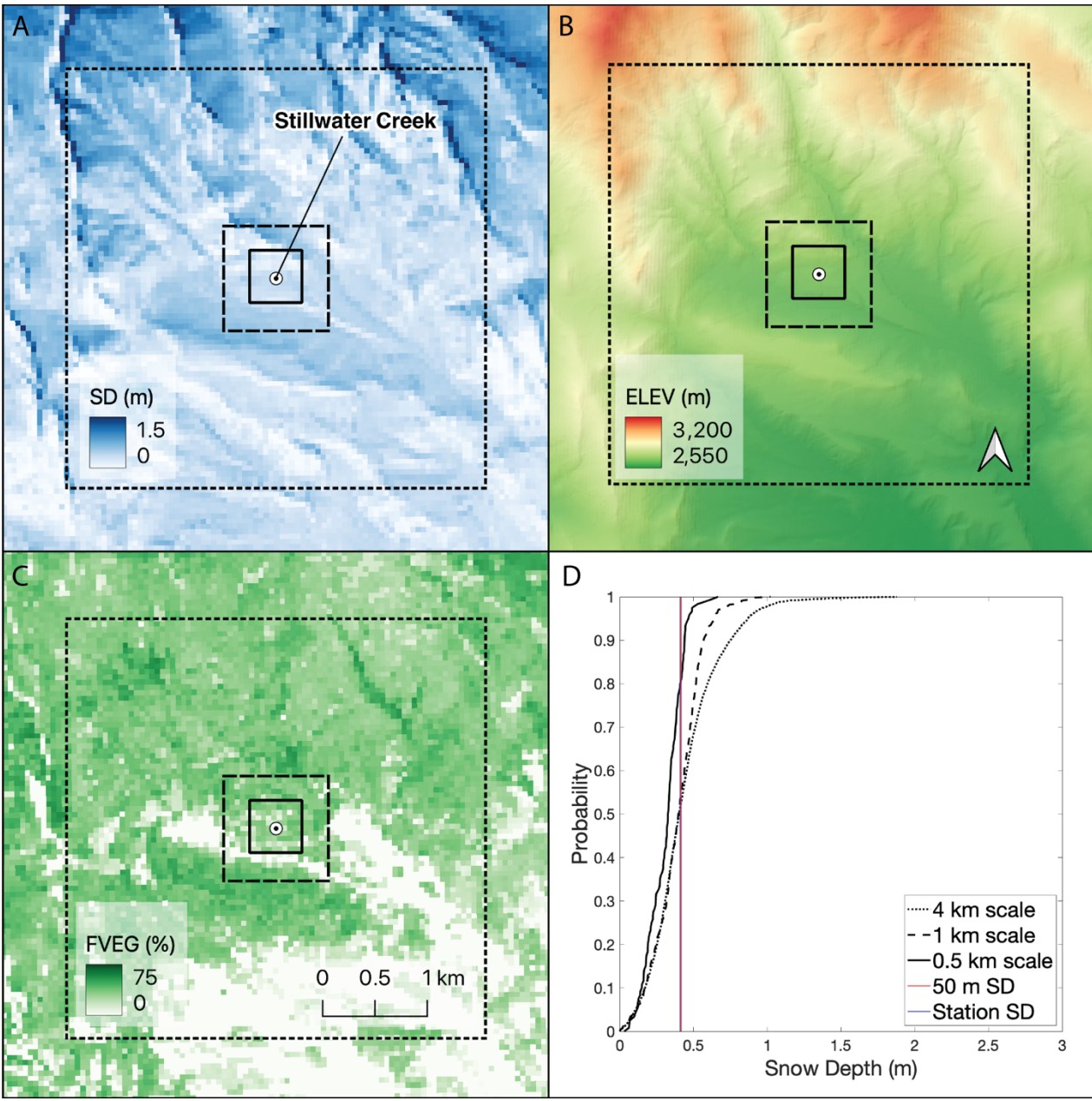


**Figure 2: The spatial distribution at 50 m resolution of (a) Lidar snow depth, (b), elevation, and (c) fractional vegetation. The squares represent spatial scales of 0.5 km (solid), 1 km (dashed), and 4 km (dotted). (d) Cumulative density functions (CDFs) of snow depth at each of the three scales with 50 m SD and station SD plotted on the distribution for the Stillwater Creek snow station in Colorado, April 16, 2023.**

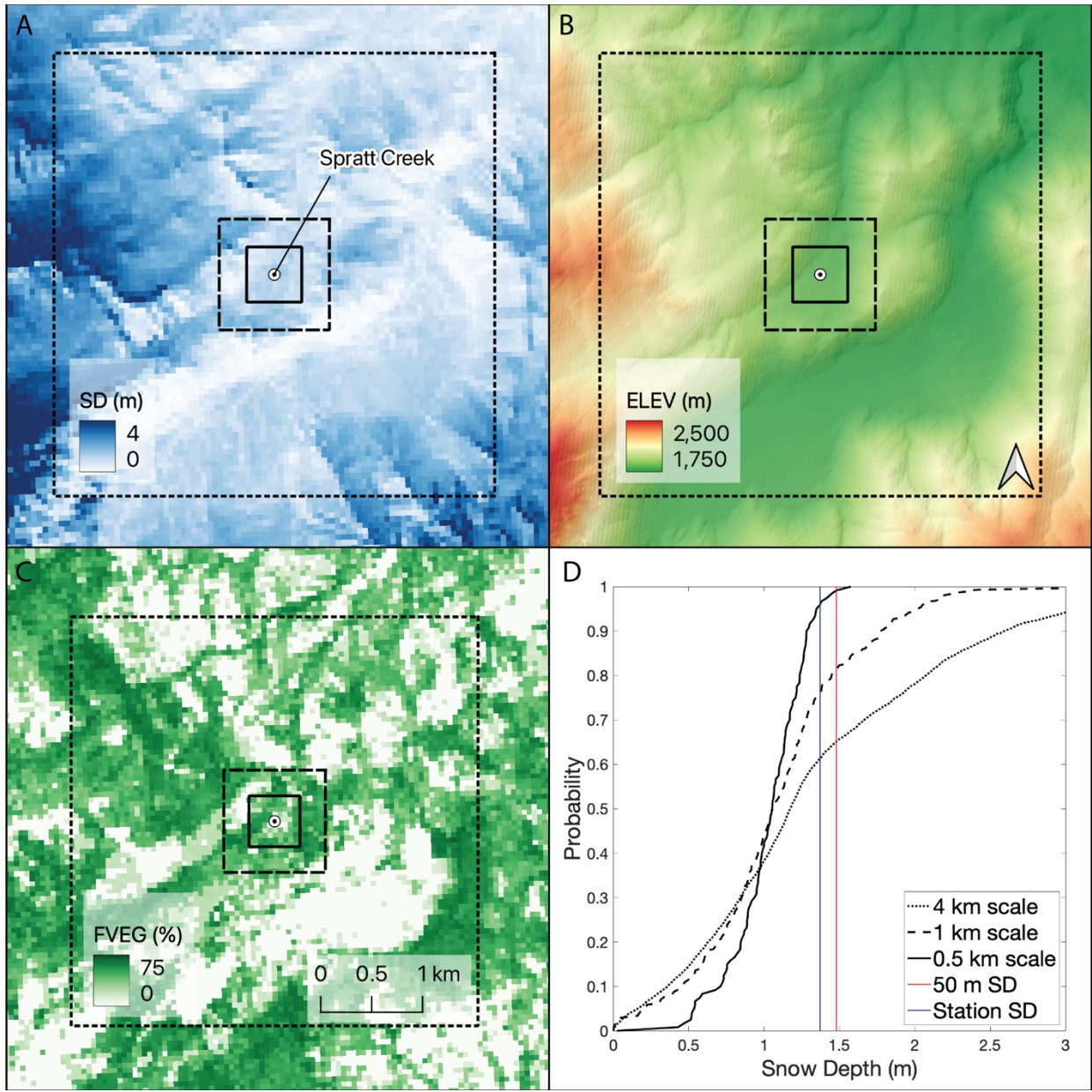


**Figure 3: The spatial distribution at 50 m resolution of (a) Lidar snow depth, (b), elevation, and (c) fractional vegetation. The squares represent spatial scales of 0.5 km (solid), 1 km (dashed), and 4 km (dotted). (d) Cumulative density functions (CDFs) of snow depth at each of the three scales with 50 m SD and station SD plotted on the distribution for the Spratt Creek snow station in California, March 31, 2023. Note that the x-axis in (d) is cut-off and that there are**

**snow depth values exceeding 3 m at the 4 km scale.**

## 2.2 Analyses

In this section, we describe the analyses conducted. First, we present the spatial scales at which we conduct the analyses, then we provide details on each analysis in the order of the research questions it aims to address.

### 2.2.1 Spatial scales

We conduct our analyses at three spatial scales typically employed in remote sensing and modeling applications: 0.5 km × 0.5 km, 1 km × 1 km, and 4 km × 4 km grid squares (hereafter: 0.5 km, 1 km, and 4 km scales) (Fig. 2, 3). The snow stations were centered within these squares (as in previous studies), though we acknowledge that snow stations will rarely be centered in gridded products (remote sensing or distributed models). We separately repeated the same analyses using the 0.5 km MOD10A1F grid from the MODIS/Terra Snow Cover Daily L3 Global 500m SIN Grid data set (Riggs and Hall, 2020), and results (not shown) were not significantly changed as compared to the 0.5 km grid centered around a snow station.

We also expand on the three discrete scales to more directly assess how representativeness and the influences of landcover and topography change with scale. Beginning at the point scale, we expand outward in 50-meter increments up to the 8 km scale. In doing so, we are able to assess the relationship of scale and representativeness as well as determine if the trends we observe continue beyond the 4 km scale. At some sites, expanding the analysis to scales greater than 4 km results in an analysis area that extends beyond the bounds of the Lidar scan. For the expanded scale analysis, we only included sites in which 90% or more of the grid cells contain snow depth values at the 8 km scale. This reduced the number of snow stations in the analysis to 56 (from 111) but ensured that the results were not influenced by increased amounts of null data at larger scales.

### 2.2.1 Snow depth variability

To gauge snow depth variability surrounding a snow station we evaluate the distribution of snow depths at each scale. To do so, we calculate the $5^{th}$–$95^{th}$ percentile range of snow depth values using the 50 m resolution Lidar data at each coincident Lidar-station pair (Fig. 2d, 3d). We then determine where point snow depth observations (station SD and 50 m SD) fall within the cumulative density function (CDF) of 50 m snow depths at each scale. We present the results of this analysis in Section 3.1

### 2.2.2 Relative snow depth and representativeness

We assess the spatial representativeness of a snow station by comparing point snow depth to the areal-mean snow depth. To do so, we employ a metric: relative snow depth (RSD). RSD is calculated by subtracting the areal-mean snow depth from the point snow depth representing the snow station, following Eq. (1):

$$RSD = point\ SD - areal\text{-}mean\ SD \tag{1}$$

We use RSD to determine if extrapolation of the point snow depth to the larger area would overestimate (if positive) or underestimate (if negative) the areal-mean snow depth. We calculate the RSD for each spatial scale, using station SD and 50 m SD as point data sources. We deem a site to be representative if the RSD is within ±10 cm. We acknowledge that the range of "representative" RSD values varies based on the application and there is subjectivity in what constitutes a representative site (similarly discussed in Meromy et al., 2013). Our results could easily be adjusted using a different range of acceptable values. We present a probability density function in Section 3.2 to illustrate the distribution of RSD values irrespective of our classification of representativeness. Unlike previous investigations, we do not use a percent difference from the mean as an indicator of representativeness, as percentages can be overly influenced by the magnitude of snow depth. The data we employ encompass a wide variety of locations and times within the snow season, meaning snow depth magnitudes are highly variable. As such, the magnitude difference is a more interpretable metric.

Snow stations are strategically placed on the landscape to maximize their utility for water supply forecasts (NRCS, 2011). We assess the impact of this strategic placement by calculating RSD for all possible snow station locations at each study site. Using Lidar data, we calculate the RSD value by sequentially setting each pixel in a study area as the snow station location. For example, we calculate 100 RSD values at the 0.5 km scale for the 100 pixels (each 50 m resolution) within the study area. We use these data to create a distribution of expected RSD values at a given scale (term: virtual RSD). We then compare the distribution of the virtual RSD values to the distribution of real RSD values (across all 476 station-Lidar survey pairings) to see how strategic placement of snow stations compares to expected RSD values. The results of these analyses are presented in Section 3.2.

### 2.2.3 Consistency of RSD values

Is the sign and magnitude of RSD at a site consistent through time? We address this question by calculating RSD at each snow station over all available Lidar surveys in the three-year period. For this temporal consistency assessment, we include all sites that have data points spanning at least three Lidar surveys across at least two years (n = 71 sites). To assess temporal consistency at snow stations, we partition the data into three groups: those where the median RSD is less than -0.1 m, between -0.1 to 0.1, and greater than 0.1 m. We then analyze the distribution of RSD values within these three groups. Additionally, we assess how RSD varies throughout the season by plotting RSD against days to snow station melt out date for each site.

### 2.2.4 Landcover and topography analysis

We assess variations in landcover and topography to test whether there are any discernable effects on RSD (Fig. 2b-c, 3b-c). To do so, we calculated relative elevation, relative fractional vegetation (FVEG), and relative southness. These metrics are similar to RSD; they are calculated by subtracting the areal-mean value of the variable from the pixel value closest to the snow station. For example, a positive relative fractional vegetation value signifies that the fractional vegetation value representing

the snow station is greater than the mean fractional vegetation of the surrounding area. We use linear regressions to determine if there are significant relationships between the relative landcover/topography variables and RSD.

## 3 Results

### 3.1 Snow depth variability

The spatial variability of snow depth influences the likelihood that a snow station is representative of the surrounding area. A higher range of snow depths increases the maximum possible magnitude of RSD, whereas a limited snow depth range has a smaller maximum RSD. For example, a site with a 20 cm range of snow depths would have a maximum RSD value of 10 cm (assuming a normal distribution), guaranteeing the station to be representative. Recall that we define a representative site as being within 10 cm of the areal-mean. Here, we examine the statistical distribution of snow depths surrounding snow stations and its role on site representativeness, with a focus on the 0.5 km scale.

The 5–95$^{th}$ percentile range of snow depth varies greatly between sites and between study region (Colorado vs. California, Fig. 4a, f). The mode for the 5–95$^{th}$ percentile range is 0.4–0.5 m in Colorado and between 0–0.1 m in California; the latter a result of Lidar surveys occurring when some study sites were mostly snow-free. Aside from these low values, most sites have a range of snow depths between 0.3–0.6 m at the 0.5 km scale in both Colorado and California. The maximum 5–95$^{th}$ percentile range is about 1 m in Colorado and 2.4 m in California, likely due to deeper snowpacks in California. The median range is 0.46 m in Colorado and 0.61 m in California.

The CDF plots demonstrate a range of possible scenarios created from different snow depth distributions. Sites characterized by lower snow depth variability (Fig. 4b, c, g, h) are less likely to have point snow depths far from the median due to the limited range of snow depths, while sites with higher snow depth variability (Fig. 4d, e, i, j) allow for greater differences between the median and point snow depth. For example, at the Michigan Creek Snotel site (Fig. 4b) the 50 m SD and station SD values correspond to the 7$^{th}$ and 95$^{th}$ percentiles, yet both values are within 0.1 m depth of the median value. Conversely, at sites with greater snow depth variability (e.g. Scotch Creek and Huysink; Fig. 4e, j), high percentiles corresponding with the station SD are accompanied by large differences from the median (0.46 m and 0.95 m, respectively). These results highlight that snow depth variability differs from site to site and that percentile from the median is influenced by the range of snow depth values. Thus, using the percentile proximity to the median is not an effective indicator of representativeness at sites with low or moderate snow depth variability. Identifying snow depth variability at sites is one important factor that controls the likelihood that a site will be representative of the surrounding area, since sites with low variability are more likely to be yield depths close to the station SD.

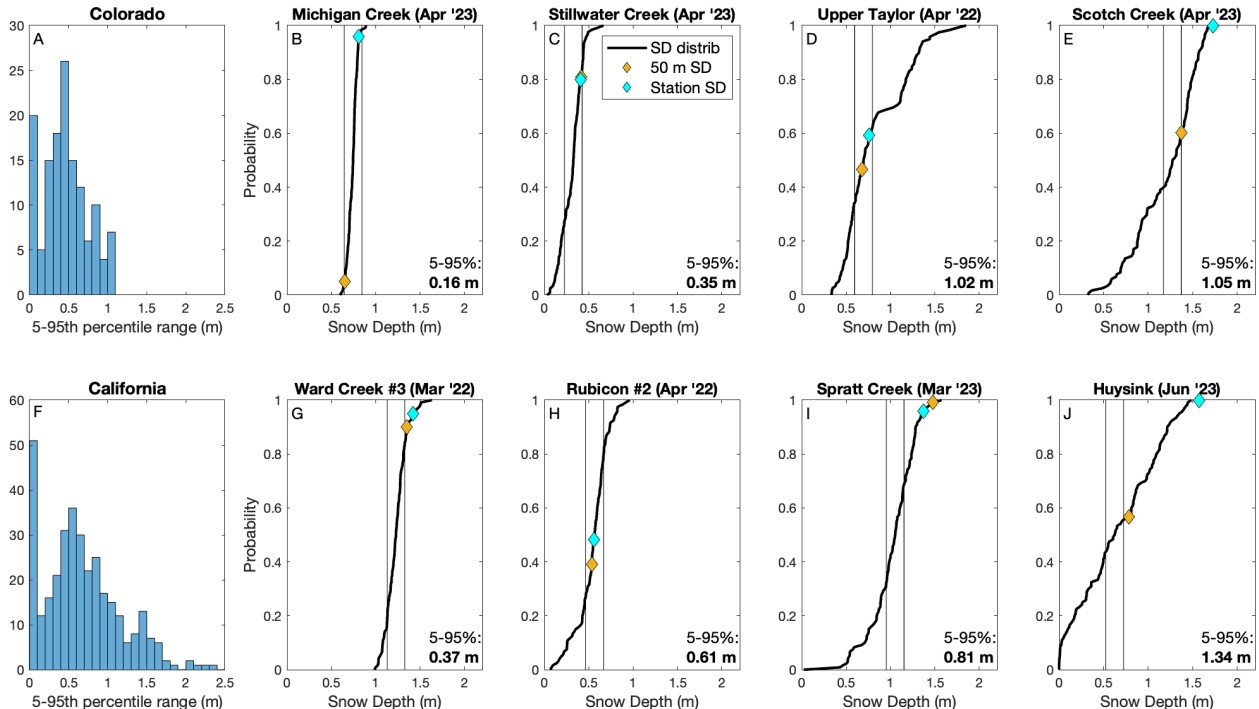

**Figure 4**: **Histogram plot of the 5–95$^{th}$ percentile Lidar snow depth values around snow stations in (a) Colorado (138 sites) and (f) California (338 sites). Cumulative density function plots at select sites in (b-e) Colorado and (g-f) California spanning low to high snow depth variability at the 0.5 km scale. Point snow depths are plotted with their corresponding probabilities within the snow depth distribution, with blue for station SD and yellow for the 50 m SD. Vertical black lines represent the range of snow depth values which are within ±10 cm of the median snow depth.**

### 3.2 Site representativeness

We now examine the distribution of RSD values, and how the distribution changes when RSD is calculated using different scales and point snow depths. This is compared to the distribution of virtual RSDs, which represent the distribution of RSDs calculated when considering each Lidar pixel in the study area to be a hypothetical station location. The virtual RSD distribution provides a distribution of RSDs if a snow station was randomly placed within the landscape.

When using station SD as the point measurement, 35%, 33%, and 28% of the snow stations are representative at the 0.5 km, 1 km, and 4 km scales, respectively (Fig. 5, Table 1). Root-mean-square error (RMSE) is 0.46, 0.48, and 0.54 m for the same respective scales. Approximately 50% of RSD values are high-biased (RSD > 0.1 m), while only ~15-21% are low-biased (RSD < -0.1 m) at all three scales.

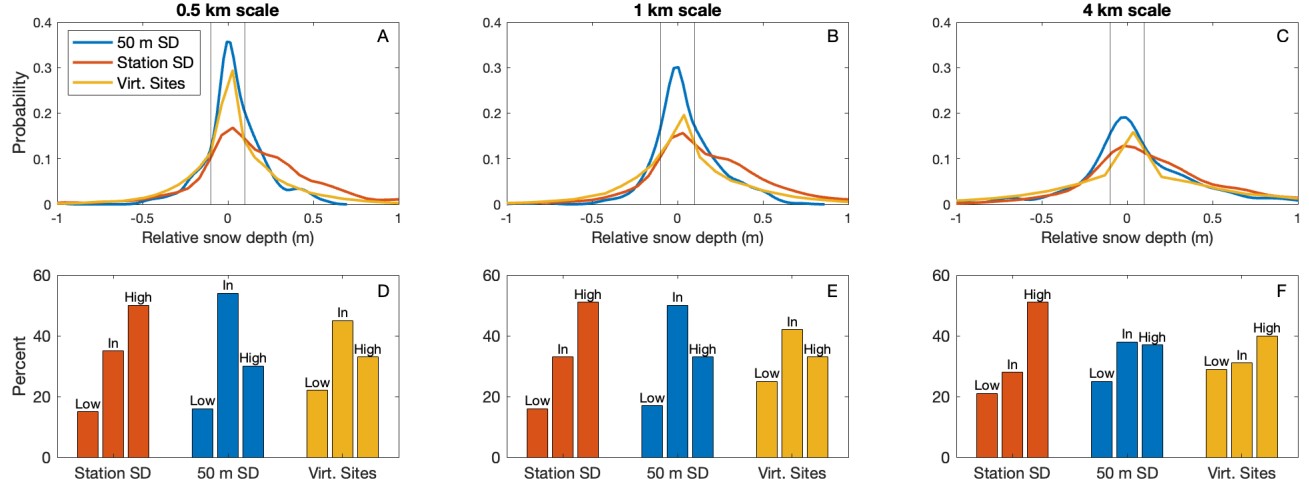

**Figure 5**: a-c) Probability density functions of RSD at the 0.5, 1, and 4 km scales, using 50 m SD, station SD, and the virtual station locations as point values for all sites. d-f) The relative distribution of RSD values which are less than -10 cm (low), within 10 cm (in), or above 10 cm (high) for each of the point values at each scale. The vertical grey lines at -0.1 m and 0.1 m represent the delineations between low-biased, representative, and high-biased sites.

**Table 1**: The percentage of coincident Lidar-snow station data points where RSD is less than -10 cm (Low), within ±10 cm (In), or above 10 cm (High) for each scale, using the 50 m Lidar, Station SD, and the virtually placed snow stations. Median, mean, and RMSE of the RSD values are also presented.

| Scale | Point Data | Low (%) | In (%) | High (%) | Median RSD (m) | Mean RSD (m) | RMSE (m) |
|---|---|---|---|---|---|---|---|
| 0.5 km | Station SD | 15 | 35 | 50 | 0.10 | 0.15 | 0.46 |
| | 50 m SD | 16 | 54 | 30 | 0.01 | 0.03 | 0.20 |
| | Virtual Site | 22 | 45 | 33 | 0.00 | 0.03 | 0.32 |
| 1 km | Station SD | 16 | 33 | 51 | 0.11 | 0.15 | 0.48 |
| | 50 m SD | 17 | 50 | 33 | 0.00 | 0.04 | 0.24 |
| | Virtual Site | 25 | 42 | 33 | 0.00 | 0.04 | 0.40 |
| 4 km | Station SD | 21 | 28 | 51 | 0.11 | 0.18 | 0.54 |
| | 50 m SD | 25 | 38 | 37 | 0.00 | 0.06 | 0.35 |
| | Virtual Site | 29 | 31 | 40 | 0.00 | 0.06 | 0.67 |

Sites are more frequently representative when using 50 m SD to represent the station (as compared to Station SD). Approximately 50% of points are representative at the 0.5 km and 1 km scales. Representativeness again decreases with scale,

with 38% of points being representative at the 4 km scale (Fig. 5, Table 1). Relative to the station SD case, RMSE values are lower when using 50 m SD, yielding values of 0.20, 0.24, 0.35 m for the 0.5 km, 1 km, and 4 km scales (Table 1). At all three scales the proportion of high-biased sites is greater than the proportion low-biased sites. Though, the difference between high and low-biased sites is less pronounced when using 50 m SD versus station SD.

The virtual snow station analysis suggests that 50 m SD locations more effectively represent the surrounding area than if they

were placed randomly (Fig. 5 and Table 1). Compared to virtual locations, real site placement (using 50 m SD) increases the frequency of representative sites and reduces the frequency of low-biased sites at all three scales. The frequency of high-biased sites is approximately equal between the 50 m SD and virtual site placement values at all three scales. We compare the 50 m SD and virtual stations to each other because they are generated from the same dataset. In doing so, the comparisons we make are a direct reflection of the location within the study area, and not any biases in sampling methodology or spatial coverage. It

is important to note that both the 50 m SD and virtual stations perform better than the station SD. We analyze the reason for decreased representativeness when using station SD in Section 3.3.

Next, we expand the spatial scales of our analysis at 50 m increments from 0.1 km to 0.8 km scales to more fully examine the effect of scale on representativeness. For both the 50 m SD and station SD the proportion of representative sites decreases with scale, plateauing at a minimum value near 20% at the ~6 km scale (Fig. 6). The main differences between the 50 m SD and

station SD results are that at the smaller scales (0.1 to 1 km) the 50 m SD values have higher proportions of representativeness, and the high bias for the station SD RSD values is consistently near 50% regardless of scale.

These results highlight that 1) point snow depths are more likely to be representative of the surrounding area at finer scales than at coarser scales, 2) non-representative sites are more likely to be biased high than biased low at all three scales and for all data sources, and 3) high-biases are most pronounced when using station SD.

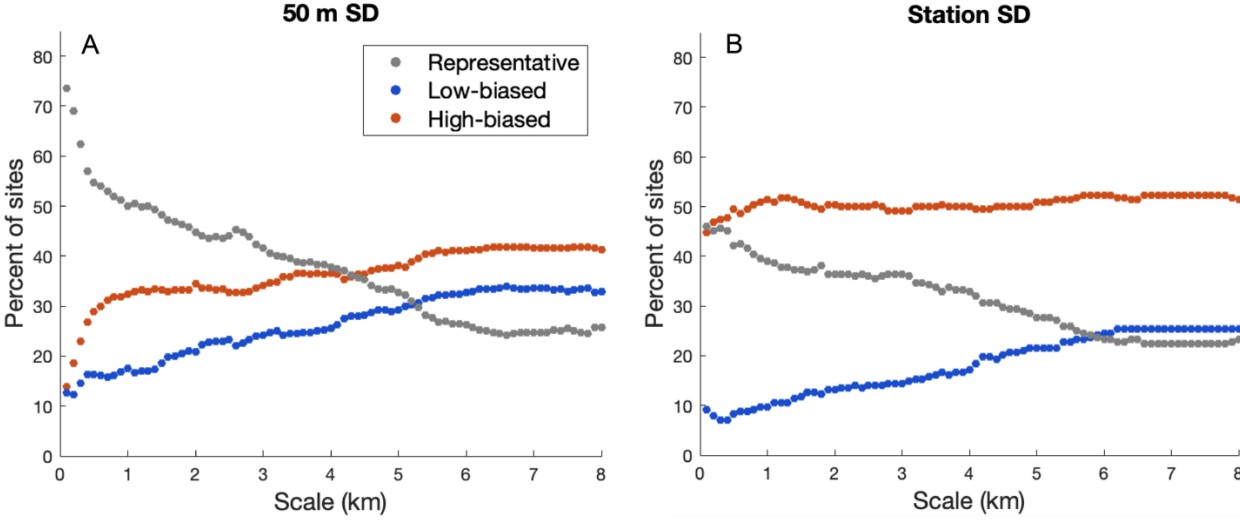


**Figure 6. The percentage of low-biased, representative, or high-biased RSD values for each scale from 0.1 to 0.8 km when using a) 50 m SD as the point value or b) station SD as the point value.**

### 3.3 Point snow depth comparisons

As exemplified in Figure 5, the source and spatial coverage of point snow depth observations influences whether a site qualifies
as representative. RSD calculated using station SD tends to have a higher bias than RSD calculated using 50 m SD (Fig. 5). There are two possible explanations for this bias: 1) snow stations tend to be installed in locations with relatively deep snow compared to the surrounding 50 m area, or 2) there is a systematic bias caused by the difference between remotely sensed Lidar and in-situ station ultrasonic measurements of snow depth. To assess the cause of these differences we now compare the 50 m SD, the 3 m SD, and the station SD with each other (Fig. 7).

Station SDs are systematically higher than the 50 m SDs, with 48% of station SDs being over 10 cm greater than their 50 m SD counterparts and only 9% being at least 10 cm less than the 50 m SD (Fig. 7a). The station SD and 3 m SD match each other more closely (Fig. 7b); 64% of points are within ±10 cm of each other, with minimal bias. The 3 m SD to 50 m SD comparison (Fig. 7c) yields similar results as the snow station SD to 50 m SD comparison (Fig. 7a), with a similar high bias. The similarity between the 3 m SD and station SD values suggest that the high bias in RSD at stations is not caused by
differences in measurement technique (i.e., airborne Lidar vs. a ground-based ultrasonic sensor). Thus, we conclude that the high bias reported by the station SD and 3 m SD is a result of differences in snow depth at the station locations compared to the surrounding 50 m area.

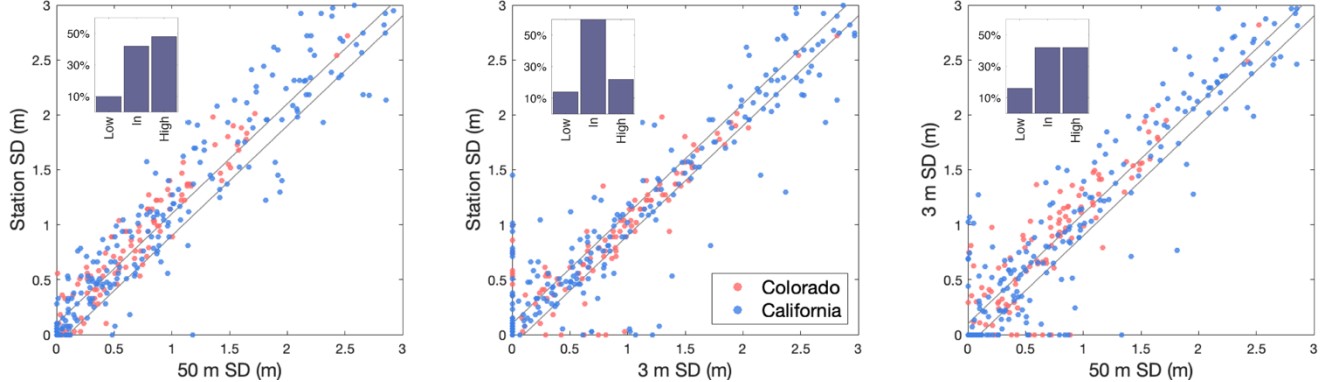

**Figure 7**: **Scatter plots comparing the three different options for point snow depth: (a) station SD versus 50 m SD, (b) station SD versus 3 m SD, and (c) 3 m SD versus 50 m SD. Points inside the black lines are within ±10 cm of each other. Histogram insets represent percentage of points that are below, within, or above the ±10 cm threshold represented by the black lines.**

### 3.4 Temporal consistency of RSD at snow stations

RSD values at individual sites demonstrate temporal consistency from survey to survey at all three scales (Fig. 8). For this analysis, we used sites with three or more lidar surveys. We grouped the sites into three categories: those with median RSD values less than -0.1 m (low-biased), between -0.1 m and 0.1 m (unbiased), or greater than 0.1 m (high-biased) at the 0.5 km, 1 km, and 4 km scales (Fig 8 a-c). Violin plots of the three categories (Fig. 8d-f) illustrate a divide between the three groups. Sites in the low-biased group are classified by almost exclusively negative RSD values whereas sites in the high-biased group are classified by almost exclusively positive RSD values. For example, at the 0.5 km scale, 64 of 65 RSD values in the low-biased group are less than or equal to zero. Similarly, 83 out of 90 RSD values are greater than or equal to zero in the high-biased group. The proportions of low and high sites are similar at the 1 km and 4 km scales. These results demonstrate that certain sites exhibit consistency in the sign of RSD values through time.

The temporal consistency of RSD at a site must be influenced by more than just relative elevation. As demonstrated in Section 3.3, the magnitude of RSD values increases in tandem with the increased magnitude of relative elevation values. However, there is still a clear temporal consistency in the sign of RSD at the smaller (0.5 km and 1 km) scales, where relative elevation has minimal influence (Fig. 8a-b). The 0.5 km scale is particularly striking; relative elevation magnitudes are generally less than 25 m (Fig. 8a), but there is still a clear delineation of low-biased and high-biased sites (Fig. 8d, f). The 4 km scale does exhibit an increased number of low and high-biased sites as well as higher magnitude RSD values, which may be a result of higher magnitude relative elevation values.

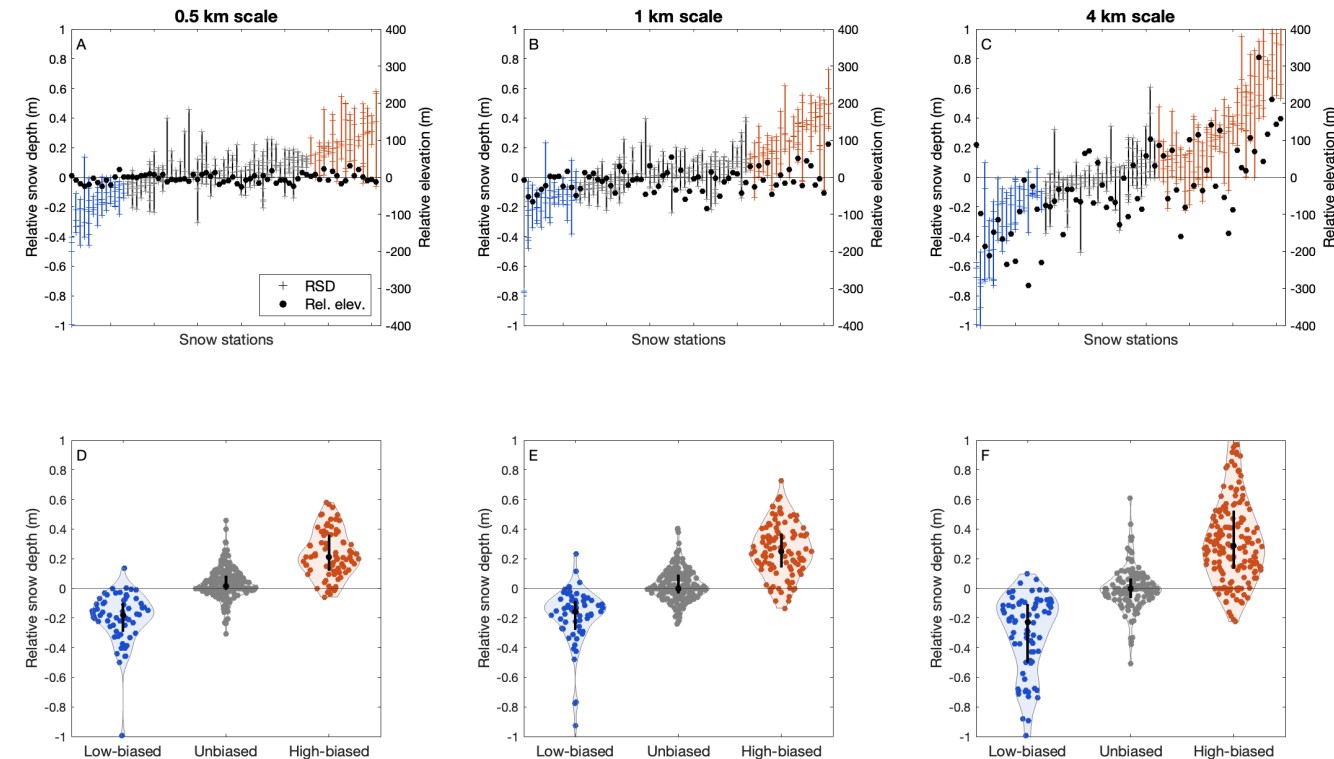

**Figure 8: The temporal consistency of relative snow depth at snow stations with three or more Lidar surveys. a-c) Each snow station (x-axis) plotted against the RSD calculated from the 50 m SD for each Lidar survey at 0.5 km, 1 km, and 4 km scales. Crosses represent individual RSD values and the lines represent the range of RSD values at a given site. Stations are ordered from lowest to highest mean RSD for each scale (thus snow stations are in different orders for each scale). Relative elevation values are also plotted as black circles on the right y-axis. d-f) Distribution plots of qualitatively grouped snow stations which are typically biased low, unbiased, or biased high for the three scales. The black bars with circles represent the median and interquartile range of the RSD values.**

The above paragraphs analyzed trends of RSD at a site regardless of timing. Here, we assess how RSD varies throughout the season. Figure 9 displays relative snow depth in relation to days from snow station melt out for three selected sites at all three spatial scales. We selected sites that yield typically negative (Devil's Postpile), variable (Dana Meadows), or positive (Ostrander Lake) RSD values. These data demonstrate that RSD does change within the snow season. At Devil's Postpile and Ostrander Lake, RSD magnitudes reach their peak in the ablation season, approximately ~50-25 days from melt out. Dana Meadows is less consistent in the timing maximum magnitude of RSD, with maximums in 2021 and 2022 occurring in the late ablation season, but the 2023 maximum occurring nearing peak snow depth. These data also suggest that scale influences the magnitude of RSD, but the sign and trend are consistent between all scales. We display three sites from California because

California sites have more Lidar surveys as well as surveys which span a greater breadth of the snow season. Colorado sites display similar trends as the sites shown in Figure 9.

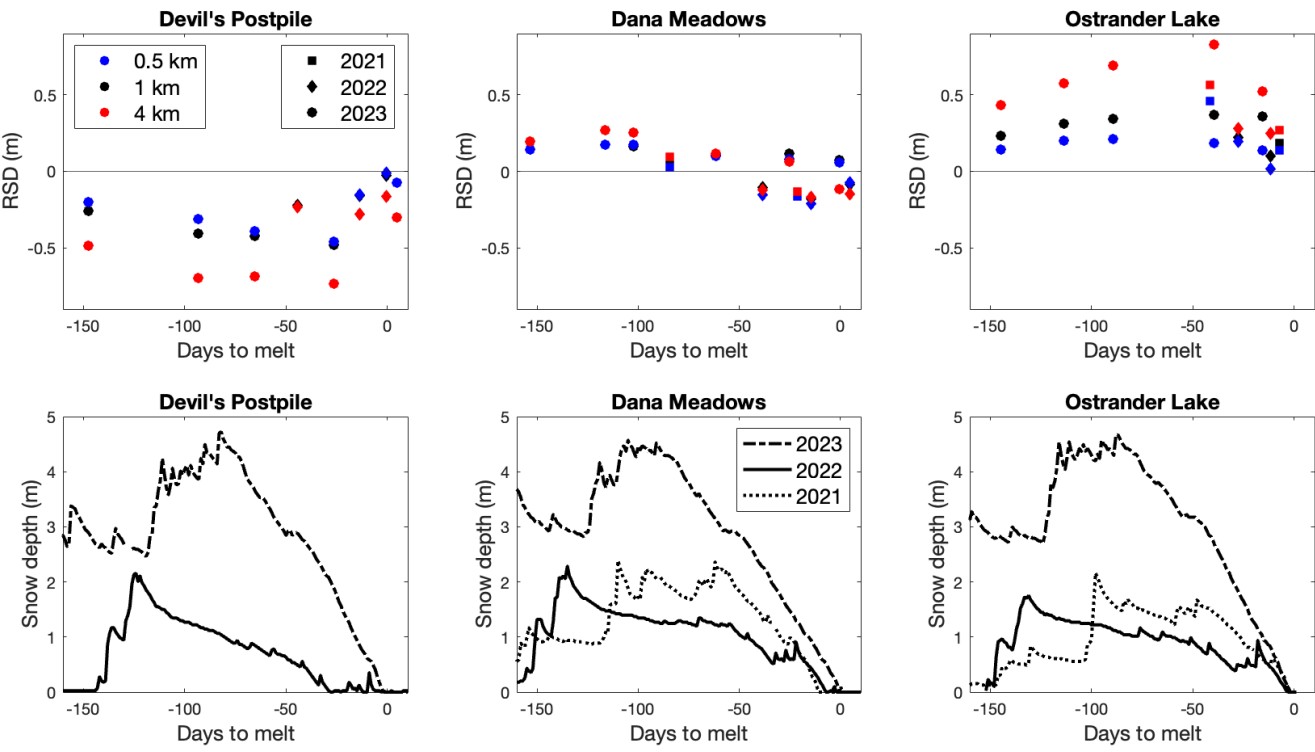

**Figure 9. Top row) Days to melt out versus relative snow depth for all Lidar surveys at select sites at all three scales. Bottom row) Snow depth time series as recorded by snow stations for years with coincident Lidar-station data at the selected site. Note that 2021 data are missing at Devil's Postpile.**

### 3.5 Topography and fractional vegetation

In this section we examine question 4: What impact do relative landcover and topography variables have on RSD? We found significant correlations between relative elevation and RSD (calculated using 50 m SD), but no significant correlations between relative fractional vegetation or relative southness and RSD. Though, regressions of fractional vegetation and southness against snow depth at each site at the 4 km scale (i.e., a regression of all 50 m Lidar snow depths values against the coincident fractional vegetation or southness value at a site) demonstrated significant relationships ($p < 0.05$) at 86% and 93% of sites for fractional vegetation and southness, respectively (results not shown). These results indicate that fractional vegetation and southness impact snow depth, however, the relative variables do not have significant correlations with relative snow depth. We discuss

possible reasons for this in Section 4.2. We focus on relationships between RSD and relative elevation hereafter, and include results related to relative fractional vegetation and relative southness in the supplement.

Analysis of the three primary scales demonstrates that the correlation (as indicated by $R^2$) between RSD and relative elevation
increases with scale (Fig. 10). At the 4 km scale, the slope of the linear regression indicates that RSD increases by 16 cm for every 100 m of relative elevation ($R^2 = 0.3$). The positive slope is consistent with our expectation of lapse rates of temperature and precipitation producing deeper snow at higher elevations.

The expanded scale analysis (0.1 km to 8 km scales) allows us to better understand the interplay of scale and elevation effects on RSD. As discussed in Section 2.2, we only include sites in which 90% or more of the grid cells contain valid snow depth
values at the 8 km scale. The correlation between RSD and relative elevation (as indicated with $R^2$) steadily increases with scale until ~7 km, where it levels off at a value of ~0.47 (Fig 11a). The relationship between RSD and relative elevation is significant ($p<0.05$) at scales greater than or equal to 0.5 km (Fig. 11a).

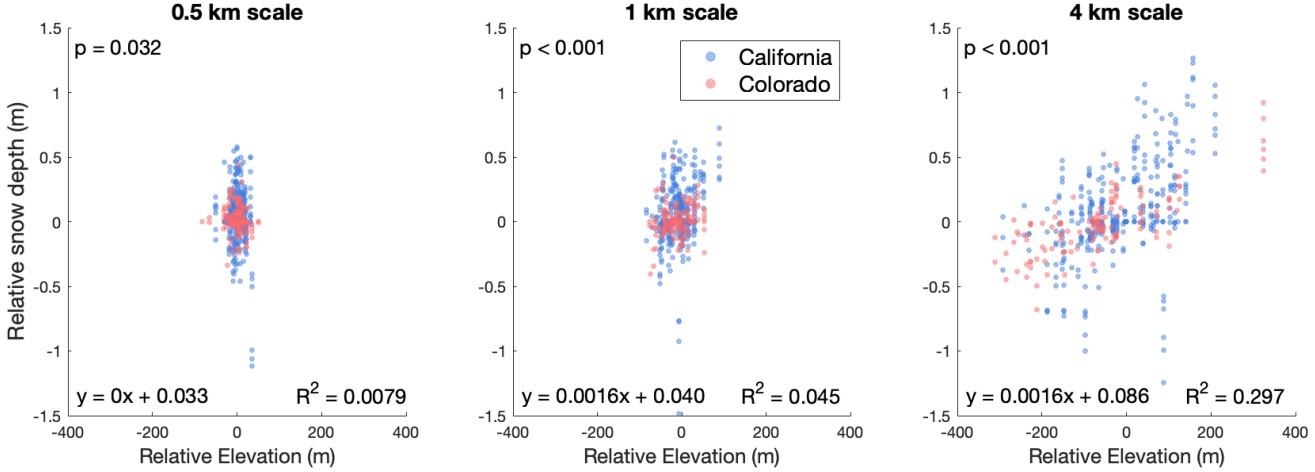

**Figure 10: Scatter plots showing the relationship between relative elevation and relative snow depth at the three spatial**
**scales using 50 m SD data to represent the point value.**

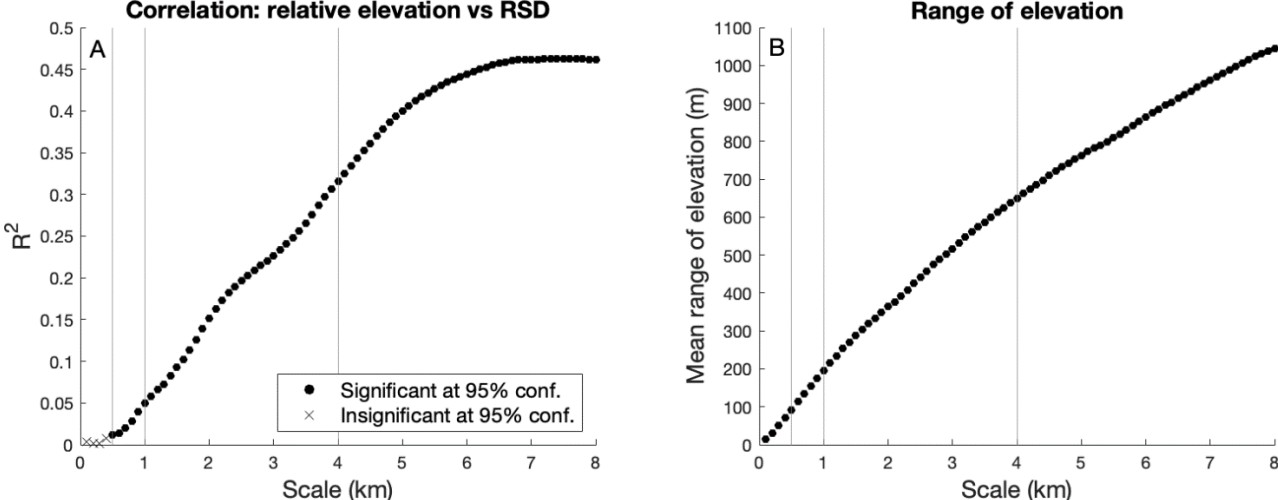

Figure 11: a) Spatial scale versus $R^2$ correlation between RSD and relative elevation. Points with p-values less than 0.05 are marked with a filled circle and sites with p-values greater than 0.05 are marked with an "x" marker. b) Scale versus the mean range of elevations calculated from all sites.

## 4 Discussion

### 4.1 High bias tendency at operational snow stations

We found that station SDs exceeded the areal-mean snow depth by at least 10 cm in ~50% of cases at all scales (Fig. 5, 6). Longer persisting snow at snow stations is beneficial for water supply forecasts, but it is unclear whether this bias is by design. The finding of snow stations to be high-biased compared to the areal-mean snow depth is not unprecedented. Grünewald et al. (2011; 2015) found that snow stations typically overestimate the mean snow depth of both the corresponding elevation band as well as the entire catchment when analyzing snow depth surrounding areas that fit the qualifications for a snow station location. Meromy et al. (2013) analyzed 53 samples, designating a site representative if the station SD was within ±10% of the areal-mean SD. Using that definition, 51% of their station SDs were representative, 30% were high, and 19% were low at the 1 km scale. This distribution more closely matches the distribution we observed when using 50 m SD as the point snow depth, but still demonstrates a slight high bias. It is important to note that the use of percentage from the areal-mean snow depth is different than our use of magnitude from the areal-mean snow depth, which could affect the results.

Comparing the snow depths we use to represent the snow stations demonstrates that the station SD values are consistently higher than the 50 m SD values (Fig. 7). The general agreement between the 3 m SD and station SD values, two independent data sources, suggests that the deeper snow depths at the snow stations are not a result of differences in sampling methodology (i.e., Lidar vs. ultrasonic depth sensor), but rather fine scale (several m) spatial variability within the 50 m pixel. A higher

proportion of sites are representative of larger areas when using 50 m SD as opposed to station SD (Fig. 5). This suggests that the high bias at the fine-scale station location lowers representativeness. Uniformly correcting the bias exhibited by snow station snow depths would mitigate this problem at some sites, but risks deteriorating representativeness at low-biased sites. Thus, bias correction would have to be site specific and require existing spatial snow depth data.

Why are station SDs higher than the corresponding 50 m SD values? There are multiple possibilities of why station location within a 50 m pixel causes a high-bias: (1) there is a persistent bias caused by snow station location or (2) the bias is caused by the snow station infrastructure. Grünewald et al. (2011) suggested that deeper snow at stations compared to the surrounding area were a result of flat terrain at a snow station compared to the sloping terrain characteristic of a mountain watershed. Persistent shielding effects or placement within forest gaps could provide another location-based explanation for the high bias.

The bias could also be introduced by the snow pillow, which is a flat, vegetation-free structure with thermal properties distinct from the surrounding forest floor. A final explanation could be that snow density is systematically lower at the snow station, so the increased SD would not actually result in differences in SWE. Density could be lower due to altered thermal exchange at the snow-ground interface due to the snow pillow (i.e., hence changing metamorphism) or due to wind sheltering (e.g., reduced rates of settlement and compaction of newer snow). This final issue highlights the limitations of working in terms of

snow depth, since spatial variations in density can influence snow depth variations (e.g., Bonnell et al., 2023; Meehan et al., 2023). Knowledge of both depth and density are needed to accurately resolve spatial distributions of SWE. In all, further work is required to ascertain the exact cause of higher snow depths recorded at snow stations compared to the surrounding 50 m area.

**4.2 Temporal consistency of station biases**

Snow stations exhibit both intra and inter-annual consistency in the directional bias of RSD. At least half of sites with three or more Lidar surveys demonstrate almost exclusively unidirectional bias in RSD at all three scales (Fig. 8). Meromy et al. (2013) also found consistent bias direction and magnitude at many sites in their investigation. Another study analyzing basin-wide snowpack using Lidar data found consistent patterns of snowpack in years with similar meteorological characteristics (Pflug and Lundquist, 2020). Topography, landcover, and typical storm tracks are relatively static on annual timescales (e.g., Liston,

1999). If these are the factors that control snow depth distribution it is not unexpected that RSD biases would also be similar from year to year at a given site.

Given this consistency, it may only take a few Lidar surveys at a site to determine the relationship of a snow station to the surrounding area at a certain scale. However, the timing of Lidar surveys within the snow season would need to be considered since the magnitude of RSD varies throughout the season (Fig. 9). Lidar survey timing is currently biased towards peak SWE

and the ablation season, with limited surveys during the accumulation season. Regardless, previous efforts to determine the relationship between a snow station and the surrounding area required labor intensive manual sampling of snow depth

surrounding a snow station. Thus, we can increase the utility of the temporally continuous snow station data with just a few Lidar surveys. The consistency we observe provides the opportunity to adjust snow station data based on the typical RSD bias at a site for other applications. Doing so would cause the adjusted value to be more in line with the areal-mean snow depth, improving its utility for remote sensing ground-truthing, data assimilation, or model validation efforts.

## 4.3 Influence of landcover and topography

Vegetation and topography influence the distribution of snow across the landscape (Anderson et al., 2014; Clark et al., 2011; López-Moreno and Stähli, 2008; Varhola et al., 2010). Previous efforts which used statistical approaches (e.g., binary regression trees) to identify the physiographic controls on snow depth surrounding a snow station determined both elevation and fractional vegetation to be major controls on snow depth variability (Meromy et al., 2013; Molotch & Bales, 2006). Rice and Bales (2010) attributed the inability of the Gin Flat snow course and snow pillow to represent larger areas to differences in the surrounding physiography. Assessing the role of specific landscape factors on relative snow depth could inform the likelihood of a site to be representative based on the surrounding physiography.

### 4.3.1 Influence of elevation

Snow depth generally increases with elevation due to increased precipitation and colder temperatures, except at the highest altitudes where wind redistribution is more significant (Grünewald et al., 2014). We found that relative elevation and RSD have significant correlations at scales greater than or equal to 0.5 km (Fig. 11a). The increasing correlation with scale is likely linked to a growing range of elevation values (i.e., complex mountainous terrain) which have an increased impact on relative snow depth (Fig. 11b). As scale increases, sites are more likely to have higher magnitude relative elevation values, leading to higher magnitude RSD values (and fewer representative sites).

The results show that the proportion of representative sites decreases with scale until plateauing between the 6–7 km scale (Fig. 6). The close matching of the representativeness curve (Fig. 6) to the $R^2$ curve (Fig. 11a) suggests that these relationships are closely linked. Within the range of scales we assessed in the available data, the larger the scale, the less likely an individual site is to be representative (until the 7 km scale). It is unclear why the proportion of representative sites stabilizes at the 7 km scale, but one possible explanation is that other local factors controlling areal-mean snow depth keep the impact of relative elevation on RSD from increasing further. It is important to note that high magnitude relative elevation values are the primary cause for deteriorating representativeness at larger scales, not the scale itself. At the 4 km scale, relative elevation alters RSD by ~16 cm per 100-meters (Fig. 10). Thus, sites with high magnitude relative elevation values could be adjusted using this slope to better represent the areal-mean snow depth. It is important to note that the slope (change in RSD per change in relative elevation) calculated here is a mean slope of all sites used in this study. Local factors impact the rate of snow depth change with elevation, so calculating a slope of relative elevation versus RSD at an individual site would be a more accurate way to adjust RSD.

### 4.3.2 Influence of vegetation

Previous studies identified fractional vegetation as a major control on snow depth distribution (Meromy et al., 2013; Molotch and Bales, 2006). We found significant relationships between fractional vegetation and snow depth (i.e., the non-relative values) at 86% of sites (at the 4 km scale), but found no significant relationships between relative fractional vegetation and relative snow depth. This indicates that vegetation does impact snow depth, but the relative metrics we employ are unable to capture this dynamic. The relationship between vegetation and snowpack is complex and non-linear, and (depending on climate) may shift within a single snow season (e.g., less deep snow in the forest in mid-winter but deeper snow in the forest in the spring melt season) (Dickerson-Lange et al., 2021; Lundquist et al., 2013; Mazzotti et al., 2020; Bonner et al., 2023). Additionally, there may be different snow depth regimes within subcanopy zones and gaps in a forest (e.g., Currier & Lundquist, 2018). Given these factors, the relationship between fractional vegetation and snow depth is much more complex than the comparatively simple (and linear) lapse rate effects of elevation on temperature and precipitation.

Accurately simulating forest effects on snow cover also requires extremely high spatial resolutions (<5 m) (Clark et al., 2011; Mazzotti et al., 2021), which would not be captured by the 30 m fractional vegetation dataset we employ. Additionally, we used relative fractional vegetation as the metric to describe site vegetation, which reduces vegetation dynamics to a single value. A single value may be insufficient to capture the complex dynamics of vegetation effects on snow. For example, an areal-mean fractional vegetation of 0.5 could represent either an area split into equal parts of 100% and 0% vegetation cover, or a homogeneous area with 50% vegetation cover. The impact of vegetation on snow distribution at these two example sites could be considerably different, but the areal-mean value is unable to convey the difference in vegetation distribution between the sites. An analysis of the high-resolution spatial distribution of vegetation involving the distribution of forest gaps would conceivably reveal the influence of vegetation on relative snow depth, but is beyond the scope of this paper.

### 4.3.4 Influence of southness

It is well documented that slope and aspect impact snow distribution (e.g., Golding and Swanson, 1986; Murray and Buttle, 2003). We similarly found significant relationships between southness and snow depth at 93% of sites (at the 4 km scale), but no significant relationships between relative southness and relative snow depth. One explanation for the lack of significant relationship is that snow station southness is not different enough from the surrounding area to impact snow depth. Snow stations are strategically placed on flat areas, which could reduce the influence of relative southness. It is possible that other landscape factors outweigh the impact of southness on snow depth, making its impact more difficult to ascertain. More complex analyses which take multiple variables into account may be required to determine the relative importance of landscape variables on relative snow depth.

## 5 Conclusions

We analyzed snow depth distributions surrounding snow stations at three scales using coincident Lidar-snow station data in Colorado and California from 2021–2023. Snow stations (Station SDs) record snow depths within ±10 cm of the areal-mean snow depth in approximately one third of cases at all three scales, while overestimating the areal-mean snow depth by greater than 10 cm in ~50% of cases. When relative snow depth is calculated using 50 m SD, the frequency of site representation is increased to ~50% at the 0.5 and 1 km scales. Representativeness increases when using 50 m SD because snow station locations record snow depths which are on average ~10 cm greater than the surrounding 50 m area. This high bias needs to be considered when using snow station data for validation. Representativeness decreases with scale because relative elevation magnitudes increase, causing lapse rates to impact relative snow depth via changes in areal-mean snow depth. The directional bias of RSD at a snow station is consistent from survey to survey. Together, these results suggest there is an opportunity to increase the utility of snow stations for model validation and ground truthing. Future work should focus on determining the underlying influences that cause site bias, potentially allowing for *a priori* identification of a site's relationship to the surrounding area. Adjusting snow station data based on the consistent high bias compared to the surrounding 50 m area, or based on the typical trend of RSD would increase the ability of a snow station to better represent the surrounding area, particularly at scales of 1 km or less.

## Code Availability

## Data Availability

Landcover and topography data were provided by the United States Geological Survey. Snow station data were provided by the USDA NRCS and the CA-DWR (https://www.wcc.nrcs.usda.gov/snow/, https://cdec.water.ca.gov/snow/current/snow/). Lidar data were provided by ASO, Inc. (https://www.airbornesnowobservatories.com/).

## Author contribution

Herbert carried out the analyses, created the figures, and wrote the manuscript. Small and Raleigh helped design the experiments as well as edit the figures and manuscript text.

## Competing interests

The authors declare that they have no conflict of interest.

## Acknowledgements

This work was supported by the National Aeronautics and Space Administration (NASA) under three grants: Water Resources Program Grant No. 80NSSC22K0928 and Terrestrial Hydrology Program Grants No. 80NSSC22K0685 and NNX17AL41G.
We would like to thank the institutions which provided invaluable data for this investigation.

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
