# Peer review of "Reanalyzing the spatial representativeness of snow depth at automated monitoring stations using airborne Lidar data"

_EGUsphere, 2023_

## Referee Comment (RC1)

Reviewer: Hannah Besso

**General Comments:**
The paper constitutes an important contribution to the field. Snow station data are used for many applications in hydrology. This study adds to our understanding of these stations' representativeness of basin snow quantities and is an important addition to snow hydrology. The scale of the analysis and use of lidar sets it apart from previous studies. However, the authors should explain better and/or reevaluate the temporal analysis. They should also remove the landcover component from Research Question 3, since the author states in the discussion that the dataset used for this component of the analysis was inadequate. Additionally, the manuscript (especially the Analyses section) should be reorganized or condensed to make the story clearer.

**Organization:** The Introduction includes a deep dive into several relevant papers, whose details are repeated later in the paper. These details should be removed from the Intro. The final paragraphs of the Intro, starting at line 91, would fit better in Methods. The Analysis section seemed to jump from one thing to the next and was confusing to keep track of what you were doing. Either introduce the whole section with a list (could even be bullet points) or make separate section headers with titles that describe what you do for each of these paragraphs/separate analyses. Then in the Results it wasn't always clear which part of the Analysis you were reporting on. It would probably be best to maintain the order in both sections, consistent with the order of the Research Questions.

**Vegetation Impacts:** The lack of a strong vegetation component is a big missing piece of this paper, and it should be highlighted as future work that should be done. I think the paper still stands without a veg component, because the impact of the paper is the bias-correction of the stations, not the reasoning behind that bias. However, understanding the impacts of vegetation on snow quantities at snow stations relative to the surrounding basins is important. Somewhere near the beginning of the manuscript you should acknowledge that vegetation has been proven to impact snow depth, but that your dataset was too coarse to adequately investigate the impacts. Or, if you want to include a vegetation component, you could come up with a simple metric such as distance from station to canopy edge (even just using imagery like on Google Earth). As the manuscript stands currently, the discussion provides good citations of others' work on snow-vegetation interactions, but your Results section reads as if you think there is no impact of vegetation on snow.

**Temporal component of the analysis:** How did you decide on the different groups (that were "typically" low, unbiased, or high)? I'm not fully convinced that these groups are distinct since there's so much overlap in Fig 9D-F.
Pflug and Lundquist (2020) (see Figure 6 of that paper) show that basin snow variability can change throughout a season based on snow covered area and whether it was a 'big' snow year or not. This seems relevant to your Section 3.1, where you argue that a larger range of snow depths increases the maximum magnitude of the RSD. So it would follow that there might be an inter- and intra-season temporal component to changes in RSD at a basin. And Figure 9A-C does show that stations can have a range of RSDs of up to about 50 cm, which seems large

relative to your 10cm threshold. I don't find myself convinced that stations are so temporally consistent in their bias that it would be easy to bias-correct them based on just a few lidar flights. I think this would be a huge finding that would have implications for everyone who uses snow station data - I just want to see this proven/investigated a bit more thoroughly. A discussion also might be warranted of whether 3 years of data is enough to develop this relationship.

**Snow depth instead of SWE:** I've been told by people more familiar with CA data than I that snow depth measurements (especially at sites in California, managed by CDEC) are less accurate because they're not maintained or quality controlled as well as the SWE measurements are. I think your reason for using snow depth is valid (given that it's directly measured by the lidar data) but it's worth thinking about and maybe mentioning more than the brief description of your quality control method.

**Technical Corrections:**
Line 65: Define 'area-mean snow depth' since you use it throughout the paper.

Line 134: "provides no advantage" - higher accuracy of SWE vs snow depth measurements from CA snow stations

Lines 129 - 132: repetitive with the intro. I think you should remove these details from the Intro.

Lines 150 - 152: Might deserve further discussion.

Line157: "requires much less storage and computational expense to manage" in comparison, I assume, to the 3 m data set instead of the "range of larger scales" you reference in the previous sentence. Be explicit here.

Lines 154 - 159: How do they produce the 50 m product? Is it derived from the 3 m product?

Lines 161 - 164: DTMs can vary quite a bit in their RMSE, and errors can be spatially variable. For example, areas of a certain DTM with steep slopes or dense vegetation can have larger errors or even systematic bias than in areas that are flat with less vegetation. I think here you could just get away with reporting the error published in Painter et al., 2016. Especially because this is the only lidar dataset you use, so I don't see any added benefit to making generalizations about other lidar products.

182 - 183: Confusing sentence, I don't understand what you did.

Line 184 - 185: Reword this sentence to be more clear. "We compared the snow depth from different data sources at each point location" or something. "Use different data sources to represent the point snow depth value" is confusing.

Line 185 - 187: Can you explicitly define SD? I understand the sentence but had to read it twice to make sure SD was defined.

Line 187 - 189: Better suited when you introduce the datasets in 2.1.3.

Paragraph starting with 192: Equation 1 and surrounding text would fit better here than in the introduction.

Line 195: Why did you choose 10 cm? Also, "acceptable" is undefined. Why not say something like "this threshold will change based on the application, but here's why we chose 10cm".

Lines 212-213: Confusing sentence and probably unnecessary. This section would likely benefit from a summary sentence at the beginning or end that lists your analyses (see my above comments on organization).

Line 215: "proportion of representative sites" was confusing, had to read twice. Make it more clear what you're referring to (I assume it refers to 2 paragraphs previous, where you talk about using each cell as the "station" location.)

Figure 2: I like that you show the different scales using the boxes. But I want the boxes to be different colors than the SD and Elev scales. Also make the SD and Elev gradients different color scales.
For D can you plot the 50m SD and Station SD as vertical lines that intersect your different CDFs instead of points? Otherwise there are 3 points on this graph representing the same data.

LIne 229: Typo: extra d in "Dd Cumulative"

Figure 3: Same critiques as Fig 2.

Line 233: You use Cumulative Density Function in Figure 2 caption but CDF in Figure 3 caption. Be consistent.

Line 234: What do you mean by "truncated"? Be explicit.

Line 245: Why include lidar flights that occurred when the study sites were mostly snow-free if this will skew your statistics?

Note: you already defined a threshold of 10 cm magnitude RSD. Why so much emphasis on percentiles? This seems like a useful tool in characterizing site variability, but you say it yourself that it's a problematic indicator of representativeness, so emphasize it less. Also, how does the timing of the lidar flights play into this quantile analysis see above comments about the temporal analysis? Do periods of ablation change this relationship?

Line 264: stick to cm units for consistency

Figure 4: Are the vertical lines on your CDFs supposed to represent the 5th and 95th percentile? They don't look like they do (they're all the same width just located in different places - maybe check your code for generating these). If they don't represent those quantiles, I think they need to be labeled/explained.

Line 289: what are "low sites"? Do you mean "low-biased sites"?

Lines 291 - 294: I like this summary at the end of the section.

Figure 5: Explain what the gray vertical lines are.

Lines 305 - 308: See my above comments on the vegetation component.

Lines 313-314: perhaps due to vegetation effects.

Figure 6: The different colors overlap such that they block each other. Is there a way to make both visible via a different type of plot or by using transparency? This is especially a problem at the 0.5 km scale where I think the pink is plotted on top of the blue. Also why is the .5 vertical line lighter than the others? I missed it at first.

Line 349: "sensing scale"? Does this refer to remote sensing or something else?

Figure 8 caption: use consistent labels. "50 m Lidar pixel" vs. "50 m SD". Also, the 10 cm lines look gray to me instead of black.

Figure 9: See my above comment about how you grouped the stations. There's a lot of overlap between groups (D-F).

Line 384: I don't think "overrepresent" is the right word here.

Lines 386 - 388: this fits better here than in the Intro.

Lines 397 - 398: Rephrase. I think you're saying that any bias correction would need to be site specific. And do you mean "positively biased" not "oversampling"?

Line 400: Instead of a list you should present the infrastructure, flat terrain, etc as components of the location bias. Otherwise this conflicts with the other 2-component list you give in Results.

Lines 444 - 445: See above comments on vegetation component of the analysis.

Line 465: "pixel", not "point" for the lidar data

---

## Referee Comment (RC2)

**Review Summary**

Herbert et al. use continuous station data and repeat lidar data acquisitions to contextualize the representativeness of station snow depth to the surrounding areas at multiple spatial scales. Through these mixed scale snow depths, the authors additionally work to identify differences in snow depth depending on the sensor and if there are temporal patterns across at each site between each of the lidar acquisition sensors. The primary results indicate that there was no significant difference between the snow depth point measurement and the 3 m lidar snow depth at the station, however significant variability in snow depth between the point locations at the 50 m lidar areal mean were present. Generally, station snow depths area high or representative at 0.5, 1, and 4 km scales while at the 50 m scale snow depth is generally representative with some high. These results indicate that the point station snow depths are representative to the surrounding area, but stations tend to be placed in areas with greater snow depth than their surroundings.

The paper addresses a serious question that has been raised many times on the representativeness of current point measurements of snow across the US at SNOTEL or equivalent sites. These questions have become increasingly important to answer with the proliferation of modelling and remote sensing efforts that utilize the sites as a tuning parameter. The increasing availability of lidar data provides an intriguing opportunity to better define the representativeness of the sites. While the paper needs refinement prior to publishing, the methods used are appropriate and provide great insight into the stationarity of snow depth point measurements at SNOTEL and CA DWR sites.

I recommend the manuscript for publishing with major modifications, and I provide comments that are necessary to address prior to publishing.

**General Comments**

The manuscript has lots of great details and presents significant scientific findings, however the lack of structure reduces the clarity of the methods and findings. The paper would significantly benefit from additional sub headers within the methods and results sections that guide the reader through the four questions being asked, this would allow the reader to link each of the methods/results to the questions and help guide the reader through the scientific story. Additionally, while there are lots of great details in the introduction and methods, much of it is repetitive and I suggest the author carefully consider what is included and what distracts from the main points of the paper and should be cut. I recommend the authors condense significant portions of the paper but additional details are needed, specifically in portions of the methods where key data decisions and assumptions are made. The paper would also benefit from consistent use of terminology when referring to snow depth measurements at the varying scales. Finally, there are a few methods that need to be further explained or clarified prior to resubmission.

**Line Review Comments**

33-34 – Tying in the purpose for this paper at this point of the paper is confusing to the reader and would be better suited for the end of the introduction.

43-89 – It would be beneficial to the article if you revisited these paragraphs and condensed them down. There is a lot of overlap and while the content of each is useful to the reader, it could be

condensed to the most relevant information for the paper (use of SNOTEL sites as model validation, assimilation datasets, and extrapolation). Then you could point out the known flaws of using point snow data to represent larger areas and begin to tie in how lidar is a useful tool to fill this knowledge gap in the literature.

91-104 – By simplifying the earlier parts of your introduction, I think you will be better able to set up this study, why it is different and important. Leaving the reader with a clear understanding of what you plan to accomplish in the study and the questions you will be answering.

96 – It might be best to leave the equation for the RSD until the methods, so it is clearer for the reader when specifically describing your methods than having to reference the equation here.

103-104 – While I think this is a fine method to take for this research, I do think that this assumption needs to be explained further in the methods.

109-110 – I do not believe this sentence is needed here, it should be mentioned in the introduction that basin wide aerial lidar is becoming more prevalent (ASO, etc.) but it distracts from the study site and data used in the paper when included in this portion of the paper.

123-126 – Do the CA-DWR sites use the same snow depth instrumentation as SNOTEL? Add a citation.

128-137 – Refer to comment on lines 103-104.

141-149 – What is the positional accuracy of Google Earth imagery? It could be worth working with the NRCS and CA-DWR to ensure you have the correct coordinates for the station and an idea of where the depth sensor is located at each.

150-153 – Why does the SNOTEL data not also need further QC? NRCS typically only corrects the daily snow depth data (midnight). What QC methods were taken other than discarding data with a greater than 50 cm difference between hours? Why 50 cm? That seems like a large upper limit that could be tightened to a lesser number that would be more typically seen as a realistic hourly snow accumulation/ablation amount. Would it make sense to use the daily data for this work instead of the hourly data? Assuming that lidar was collected on clear sky days, can we assume that there was minimal snowfall/snowmelt during the day? Do you use the hour of the flight to compare between the station and the lidar?

174 – Additional sub headers of the different analyses would be beneficial for the reader to understand what methods you are using for each of the separate research questions.

186-187 – I think it would be beneficial to the reader if you continued to call these three point measurements by the names listed here throughout the paper.

187-188 – This line about 50 m resolution has been included multiple times. I recommend you remove repetitive text throughout the paper unless it is a key result that deserves repetition.

194 – The idea of identifying the representative of the stations to the surrounding area is an important topic, but I wonder if you could use a percentile-based approach that would get the user a percentage of accuracy at each of the scales that they could then judge what "acceptable" is for their use case. This approach would remove the limitations you mention in line 195, although as you mention there are limitations to percentile based analyses.

206 – For topography, why are you only analyzing elevation or is aspect also included in the topography analysis? Elevation and landcover are two key portions of a complex relationship between snow depth and mountainous terrain.

212 – This section could be trimmed down to better describe the analysis to the reader succinctly, but additional detail on what you are doing here could also be beneficial. Are you trying to identify the distance at which sites become unrepresentative to the surrounding area? I think this analysis would benefit from some more spatial statistics like variograms (Anderson et al. 2014) or assigning the percentile of representativeness to each scale. Additionally, when stretching to 8km scales, were major topographic features (i.e. ridgelines with low snow due to high winds, topographic basin directions which lead to preferential snow fall, etc.) accounted for in the analysis?

221 – Are the other analyses only conducted at one survey date, averages of each date, or are those results also temporal?

Figures 2 and 3 – Although the x-axis scales would be different between figure 2d and 3d, I do think it would be interesting to show the full cdf for both sites.

245 – Are there surveys available for periods when the stations have snowpack still? Is this the SNOTEL point data of the RSD distributed data? If RSD, at what scale? Can you drop the surveys with no snow.

261-264 – How often were the 50 m and SNOTEL measurements ~30 cm different? Keep the units the same (cm or m) when comparing two values.

Figure 4 – It might be helpful for the reader if you added a median line to the plots.

282 – A more thorough comparison of station SD and 50 m lidar SD at the same site could be very informative and help the reader better understand the differences in results between each of the RSDs and the "point" measurement. Jumping right in without this context makes the differences in RMSE hard to understand.

287, 302 – What are the virtual snow stations? What are the Sim. Sites? These all need to be called the same thing if they are, or better explain each in the methods.

305 – I really appreciate you bringing the question back to the reader, would be helpful if you did this for the other sections as well.

Figure 6 – I think it would be helpful to plot the linear trend lines that you have defined to draw the readers eye to the trends/lack of trends within each. Additionally, are blue points being blocked by pink? It would be interesting to see the difference between the states since we know Colorado and California snowpacks act very differently.

321 – Why are the sites more likely to have a higher magnitude relative elevation? Is this because sites are typically at low elevation compared to their surroundings?

335 – Figure 7c, I am not sure what this is adding, we know that SNOTEL sites are located in mountainous areas with complex terrain features, leading to significant variability (increasing STD) in elevation over larger areas.

338 – This section needs to be the first of the results. It sets the stage for all of the other analysis. Then you can identify why you use the point measurement you choose (50 m SD) to complete the remainder of the analysis. Why did you choose to use the 50 m SD for the other analysis?

Figure 8 – Adding the 1:1 line while making the +-10 cm lines darker would be very helpful for the reader. Again, transparency or making CO and CA their own plots would also be informative.

357-365 – This analysis is confusing to me and I think needs further explanation. Does the site need to be "high" at all three scales to be in the "high" grouping or are the groupings different based on each spatial scale? Why are the groups the same size, shouldn't the sites determine if they are negative/zero/positive? Are the RSDs below zero in Figure 9d-f due to the station being grouped into the three groups across all scales not for each scale individually? Line 375 starts to answer this but needs to be better included in the body of the paper.

393 – Are these completely independent data sources, or does ASO do any QC/shifting of the point cloud based on the SNOTEL site snow depth?

398 – Removing this bias systematically? Or by each site? I think this would have to occur on a site by site basis since each SNOTEL will act independently from its surroundings.

435 – Could this be a scale issue? As you mentioned, it is well documented that there is less snow under canopy than open areas depending on the time of year. Is this due to the way RSD is calculated?

465 – another name for the 50 m point measurement, please coordinate these throughout the paper to simplify for the reader.

**Technical Comments**

229 – typo "d" should be a ")"?

247 – missing units on the snow depth range

331 – Figure 7b, missing ")"

395 – "location bias."

**References**

Anderson, B. T., J. P. McNamara, H.-P. Marshall, and A. N. Flores (2014), Insights into the physical processes controlling correlations between snow distribution and terrain properties, Water Resour. Res., 50, 4545–4563, doi:10.1002/2013WR013714.

---

## Referee Comment (RC3)

Comment on RC1
Reviewer: Hannah Besso

Thank you for your reply. I think many of the changes you propose will improve the paper. I think adding the intra-seasonal variability plots at select sites is helpful for understanding the consistent sign of the bias. I like that you show one of each site with consistent negative bias, positive bias, and relatively neutral bias. My one comment on those plots is that it might be more meaningful to choose a variety of sites that are more spread out to better represent the different regions in your study. Devil's Postpile, Dana Meadows, and Ostrander Lake are all within or adjacent to Yosemite NP, so maybe consider replacing a few of those with sites in Colorado that have a similar bias signal.

My main reason for writing a follow-up comment has to do with the vegetation component of the study. I'm sorry that my communication about this component was not as clear as it could have been. I don't think you need to include details of the insignificant results, since in your discussion you say that the dataset was insufficient. Instead, you should use a dataset better able to capture the relationship between vegetation and RSD – at best, use the canopy height from the lidar data. If you don't have time to add this analysis, you should remove the vegetation component from Research Question 3, and you can still have the discussion section that talks about the importance of vegetation but that you were unable to investigate this in the analysis. I think if you keep your analysis as-is and report the insignificant results (as you do now or in greater detail), you run the risk of people mis-citing your work to say that vegetation in general has no relationship with RSD, which does not seem to be what you concluded.

Finally, it might be worth mentioning in your discussion that there is a benefit to the 'high bias' at snotel stations. Once a station melts out, we lose any information it could provide about snow in the rest of the basin. Stations placed in high elevation locations that melt out later mean that we continue to gain information for longer into the melt season, and we see from your paper and Pflug et al. (2022) that fairly consistent relationships can be developed between snow at a station and snow throughout the basin even if absolute depth is not the same. So the 'high bias' stations can be seen as providing a longer time series (and therefore more information) than can 'negative bias' stations that melt out earlier.

I am happy to continue this conversation via email or on a call if there are questions about my reply. Please don't hesitate to reach out! I'm excited to see this work published.

-Hannah
Bessoh2@uw.edu

---

## Author Comment (AC1)

**Note: author responses to reviewer comments will be written in blue text**.

We would like to sincerely thank the reviewer for their comprehensive assessment of this manuscript. We believe the alterations prompted by this review will improve the science and its presentation prior to publication. First, we summarize the substantive changes we will make to the manuscript. We respond to the line-specific comments in the second section of the document.

**Land cover and topography**

Both reviewers highlighted that our landcover and topography analysis focused on elevation and not other influences on snow distribution such as vegetation and slope / aspect. This was because we only found significant relationships between relative elevation and relative snow depth and not the other factors. We will update the manuscript to include more details about the landcover/topography analysis. We propose the following changes:

- We will add a section to the introduction discussing how the factors that control snow distribution vary with scale. The scales of our analysis are large (km scale) and thus are unable to capture all of the influences on snow distribution.

- We will add the correlation between FVEG/RSD and northness/RSD to the graph in figure 7C. While these relationships are insignificant, this figure will illustrate that we did examine topographic and landcover attributes other than elevation.

- We will expand our discussion to include more details about our expectations for landcover and topography on RSD. The influences of landcover and topography change with scale. Our hypothesis is that at kilometer scales, the effects of elevation are more pronounced than finer scale effects such as vegetation. The aggregation of a physiographic variable to a single value over a kilometer scale may hinder its utility to represent the complex dynamic of a snowpack.

**Organization**

We will condense and re-organize certain parts of the manuscript.

- The literature review presented in the introduction in the two paragraphs from lines 57-80 will be made more succinct. We will summarize the type of research that has been previously conducted in the area but will refrain describing the results in detail. Relevant results will be brought up in the discussion.

- We will remove the equation for RSD and place it within the methods section.

- We will add sub-headings to the analysis section to improve clarity and readability. We will ensure that the order of the analysis section matches the order in which we present the results.

**Temporal consistency analysis**

Both reviewers brought up the qualitative site groupings in Figure 9. We will group the figures quantitatively, with groupings based on the median RSD value. Delineations will be made between sites with mean RSD values of < -0.1m, between -0.1 and 0.1 m, and >0.1 m. An updated figure 9 is presented below:

[Figure]

Second, version 1 of the manuscript did not examine the intra-seasonal variation of relative snow depth. We will add a figure which highlights the intra-seasonal variability of RSD at three selected sites (see below). This figure will be included in the temporal analysis and be included as Figure 10 in the manuscript. This figure 1) highlights that RSD varies seasonally, peaking near peak SWE and reducing in magnitude as melt-out date approaches, and 2) it provides an additional illustration that many sites do have a consistent sign of RSD at all three scales.

[Figure]

**Review Summary**

Herbert et al. use continuous station data and repeat lidar data acquisitions to contextualize the representativeness of station snow depth to the surrounding areas at multiple spatial scales. Through these mixed scale snow depths, the authors additionally work to identify differences in snow depth depending on the sensor and if there are temporal patterns across at each site between each of the lidar acquisition sensors. The primary results indicate that there was no significant difference between the snow depth point measurement and the 3 m lidar snow depth at the station, however significant variability in snow depth between the point locations at the 50 m lidar areal mean were present. Generally, station snow depths area high or representative at 0.5, 1, and 4 km scales while at the 50 m scale snow depth is generally representative with some high.

These results indicate that the point station snow depths are representative to the surrounding area, but stations tend to be placed in areas with greater snow depth than their surroundings.

The paper addresses a serious question that has been raised many times on the representativeness of current point measurements of snow across the US at SNOTEL or equivalent sites. These questions have become increasingly important to answer with the proliferation of modelling and remote sensing efforts that utilize the sites as a tuning parameter. The increasing availability of lidar data provides an intriguing opportunity to better define the representativeness of the sites. While the paper needs refinement prior to publishing, the methods used are appropriate and provide great insight into the stationarity of snow depth point measurements at SNOTEL and CA DWR sites.

I recommend the manuscript for publishing with major modifications, and I provide comments that are necessary to address prior to publishing.

**General Comments**
The manuscript has lots of great details and presents significant scientific findings, however the lack of structure reduces the clarity of the methods and findings. The paper would significantly benefit from additional sub headers within the methods and results sections that guide the reader through the four questions being asked, this would allow the reader to link each of the methods/results to the questions and help guide the reader through the scientific story. Additionally, while there are lots of great details in the introduction and methods, much of it is repetitive and I suggest the author carefully consider what is included and what distracts from the main points of the paper and should be cut. I recommend the authors condense significant portions of the paper but additional details are needed, specifically in portions of the methods where key data decisions and assumptions are made. The paper would also benefit from consistent use of terminology when referring to snow depth measurements at the varying scales. Finally, there are a few methods that need to be further explained or clarified prior to resubmission.

**Line Review Comments**
33-34 – Tying in the purpose for this paper at this point of the paper is confusing to the reader and would be better suited for the end of the introduction.

We will delete the last sentence of this paragraph.

43-89 – It would be beneficial to the article if you revisited these paragraphs and condensed them down. There is a lot of overlap and while the content of each is useful to the reader, it could be condensed to the most relevant information for the paper (use of SNOTEL sites as model validation, assimilation datasets, and extrapolation). Then you could point out the known flaws of using point snow data to represent larger areas and begin to tie in how lidar is a useful tool to fill this knowledge gap in the literature.

We will condense this section to improve conciseness. Though, we do believe that this section provides crucial background information to the reader in that it 1) provides context for the uses of snow station data and 2) discusses the results found in previous investigations which assessed representativeness of point snow data. We will reduce the section to a general summary of the existing literature and only include key results.

91-104 – By simplifying the earlier parts of your introduction, I think you will be better able to set up this study, why it is different and important. Leaving the reader with a clear understanding of what you plan to accomplish in the study and the questions you will be answering.

96 – It might be best to leave the equation for the RSD until the methods, so it is clearer for the reader when specifically describing your methods than having to reference the equation here.

The equation for RSD will be moved to the methods.

103-104 – While I think this is a fine method to take for this research, I do think that this assumption needs to be explained further in the methods.

Regarding the decision to use snow depth and not SWE we will add "See section 2.1.2 for further explanation on this decision" At the end of the paragraph.

109-110 – I do not believe this sentence is needed here, it should be mentioned in the introduction that basin wide aerial lidar is becoming more prevalent (ASO, etc.) but it distracts from the study site and data used in the paper when included in this portion of the paper.

Will delete this sentence. The increased availability of lidar data discussion will be left to earlier in the introduction.

123-126 – Do the CA-DWR sites use the same snow depth instrumentation as SNOTEL? Add a citation.

Yes, CA-DWR depth sensors are the same as NRCS sensors based on personal communication. We have been unable to find documentation on CA-DWR snow stations. If we are unable to find any documentation we will email people at CA-DWR to get confirmation regarding the instrumentation used in DWR snow stations.

128-137 – Refer to comment on lines 103-104.

This section provides our justification for using the ASO snow depth product over the SWE product. We will add: "Lidar SWE products use modeled density (Painter et al., 2016), increasing the uncertainty of the SWE product compared to snow depth" to follow the first sentence of the paragraph. This provides an initial justification for our employment of snow depth. The following sentences describe how an assumption of uniform snow density across the landscape provides no advantage over using snow depth as the key variable.

141-149 – What is the positional accuracy of Google Earth imagery? It could be worth working with the NRCS and CA-DWR to ensure you have the correct coordinates for the station and an idea of where the depth sensor is located at each.

Lines 144 and 146 describe that Google Earth imagery is available to the fifth decimal place in decimal degrees (~1 m resolution). We utilized the official coordinates provided by the NRCS and CA-DWR (i.e., the most accurate available data). We found that the location data was not always precisely located on the snow station in Google Earth, necessitating updated coordinates in certain cases. We only used sites that we could visually confirm using satellite imagery.

150-153 – Why does the SNOTEL data not also need further QC? NRCS typically only corrects the daily snow depth data (midnight). What QC methods were taken other than discarding data with a greater than 50 cm difference between hours? Why 50 cm? That seems like a large upper limit that could be tightened to a lesser number that would be more typically seen as a realistic hourly snow accumulation/ablation amount. Would it make sense to use the daily data for this work instead of the hourly data? Assuming that lidar was collected on clear sky days, can we assume that there was minimal snowfall/snowmelt during the day? Do you use the hour of the flight to compare between the station and the lidar?

The existing text contains a mistake. We *do* employ daily snow depth data, not hourly snow depth data. We switched from hourly to daily data prior to the first submission of this manuscript and did

not update the text to reflect the change. We will update the text to specify that we use daily data for NRCS and CA-DWR sites.

Re: QC. We performed QC on all station data. NRCS stations did not require any data deletion, but CA-DWR stations did. The most common error in the data was a rapid shifts of snow depth in one direction followed by a shift in the opposite direction. The figure below illustrates an example of such a shift at ~120 on the x-axis. We found that accounting for 50 cm multi-directional shifts in snow depth reduced obvious errors in snow depth data.

[Figure]

174 – Additional sub headers of the different analyses would be beneficial for the reader to understand what methods you are using for each of the separate research questions.

We will add sub headers to this section to improve clarity / readability. Additionally, we will ensure that the order of the analysis section follows the general order of our presentation of the results section.

186-187 – I think it would be beneficial to the reader if you continued to call these three point measurements by the names listed here throughout the paper.

We will make sure to be consistent with this terminology throughout the paper.

187-188 – This line about 50 m resolution has been included multiple times. I recommend you remove repetitive text throughout the paper unless it is a key result that deserves repetition.

We will delete the justification for using 50 m data in lines 187-188. This sentence is repetitive after the justification we provide in the data section (156-159).

194 – The idea of identifying the representative of the stations to the surrounding area is an important topic, but I wonder if you could use a percentile-based approach that would get the user a percentage of accuracy at each of the scales that they could then judge what "acceptable" is for their use case. This approach would remove the limitations you mention in line 195, although as you mention there are limitations to percentile based analyses.

The limitations that we mention in line 195 are meant to showcase that an acceptable range of error is dependent on the application for which snow station data is used. This issue would persist for any metric we use to determine representativeness. Meromy et al. (2013) use an acceptable error range of 10% for RSD, but state than any cut-off for acceptability is relatively arbitrary.

In the preparation of this manuscript, we have used absolute magnitude, percent difference, and percentile (rank within the cdf of all snow depths) for classification of RSDs. Meromy et al. were able to use percentile with more success because their analyses did not include surveys near the snow melt-out date (i.e., the range snow depth values was relatively constrained). Our analysis includes data from many different sites, years, and times within the snow season, meaning we have a wide range of snow depth magnitudes (~0-6 m). This range of magnitudes makes a percent difference approach difficult to interpret between data points.

In figure 4 we demonstrate the difficulty of using a percentile-based approach. Areas with uniform snowpack may yield high percentiles for snow depths which are close in magnitude to the point snow depth (e.g., Figure 4B, lines 251-254).

Due to the complications with using percentage and percentile approaches for RSD classification, we elected to use an absolute magnitude approach for RSD classification.

206 – For topography, why are you only analyzing elevation or is aspect also included in the topography analysis? Elevation and landcover are two key portions of a complex relationship between snow depth and mountainous terrain.

We address this comment in the beginning of the document.

212 – This section could be trimmed down to better describe the analysis to the reader succinctly, but additional detail on what you are doing here could also be beneficial. Are you trying to identify the distance at which sites become unrepresentative to the surrounding area? I think this analysis would benefit from some more spatial statistics like variograms (Anderson et al. 2014) or assigning the percentile of representativeness to each scale. Additionally, when stretching to 8km scales, were major topographic features (i.e. ridgelines with low snow due to high winds, topographic basin directions which lead to preferential snow fall, etc.) accounted for in the analysis?

Yes, the purpose of this section is to assess how representativeness, as well as the influences of landcover and topography change with scale. We will edit the text where we describe the qualifications for inclusion in the analysis (lines 216-220) to be more concise.

Figure 7C shows how the standard deviation of elevation changes with scale. This is essentially a variogram conducted in grid space as opposed to the transect version presented in Anderson et al. (2014). Figure 7D shows demonstrates representativeness changes with scale.

221 – Are the other analyses only conducted at one survey date, averages of each date, or are those results also temporal?

We conduct all analyses with all available coincident lidar-snow station data. For each lidar flight in CO and CA between 2021-2023 we find all available snow station data spatially and temporally coincident with the flight and calculate relative snow depth for that date/location. We will update the text to make this clearer.

Figures 2 and 3 – Although the x-axis scales would be different between figure 2d and 3d, I do think it would be interesting to show the full cdf for both sites.

We will update the x-axis in figure 3 to include the entire cdf of snow depth.

245 – Are there surveys available for periods when the stations have snowpack still? Is this the SNOTEL point data of the RSD distributed data? If RSD, at what scale? Can you drop the surveys with no snow.

Yes, we include all available Lidar data. Some flights occurred in the late snow season, while others were flown closer to peak SD. We do not delete the low snow flights because snow melt-out at the 500 m scale may not reflect snow melt-out at the 4 km scale.

261-264 – How often were the 50 m and SNOTEL measurements ~30 cm different? Keep the units the same (cm or m) when comparing two values.

We will update the document to ensure that all units are reported in meters, not cm. We address the differences between station SD and 50 m SD in later sections (Fig 8).

Figure 4 – It might be helpful for the reader if you added a median line to the plots.

We attempted a median line in Figure 4, but the additional horizontal line makes the figure too busy.

282 – A more thorough comparison of station SD and 50 m lidar SD at the same site could be very informative and help the reader better understand the differences in results between each of the RSDs and the "point" measurement. Jumping right in without this context makes the differences in RMSE hard to understand.

We do this comparison later in the manuscript (Figure 8).

287, 302 – What are the virtual snow stations? What are the Sim. Sites? These all need to be called the same thing if they are, or better explain each in the methods.

The use of 'Sim. Site' in Table 1 was a mistake. We will update to 'virtual site' to be consistent with the rest of the text.

305 – I really appreciate you bringing the question back to the reader, would be helpful if you did this for the other sections as well.

Figure 6 – I think it would be helpful to plot the linear trend lines that you have defined to draw the readers eye to the trends/lack of trends within each. Additionally, are blue points being blocked by pink? It would be interesting to see the difference between the states since we know Colorado and California snowpacks act very differently.

We will add transparency to the markers and add trendlines to figure 6.

321 – Why are the sites more likely to have a higher magnitude relative elevation? Is this because sites are typically at low elevation compared to their surroundings?

We will add: "due to the increased range of elevations values with scale" to line 322. The reason for increased relative elevation with scale is simply because stations are located in heterogeneous terrain, meaning larger scales are likely to be more different in elevation than smaller scales.

335 – Figure 7c, I am not sure what this is adding, we know that SNOTEL sites are located in mountainous areas with complex terrain features, leading to significant variability (increasing STD) in elevation over larger areas.

The purpose of this is to demonstrate the point in the comment above: larger scales have larger ranges of elevation. This may be more clear if we change standard deviation of elevation to the 5-95th percentile of elevation in figure 7C. We are trying to show that with increasing scale 1) the range of elevations increases and 2) the correlation between relative elevation and relative snow depth increases, and 3) the reason for decreased representativeness at larger scales is likely due to increased influence of relative elevation.

338 – This section needs to be the first of the results. It sets the stage for all of the other analysis. Then you can identify why you use the point measurement you choose (50 m SD) to complete the remainder of the analysis. Why did you choose to use the 50 m SD for the other analysis?

We use 50 m SD for most of our analyses because it ensures that there is no bias caused by differences in sampling methodology (in-situ versus lidar). Comparing areal-mean lidar to point lidar allows for a direct analysis of the location of the snow station within the landscape.

We will highlight this further in the analysis section when describing the point data. Second, we will reference our point snow depth comparisons in section 3.2 to inform the reader that we do examine causes for differences in point snow depths.

Figure 8 – Adding the 1:1 line while making the +-10 cm lines darker would be very helpful for the reader. Again, transparency or making CO and CA their own plots would also be informative.

We will increase the transparency on the scatter dots and make the +/-10 cm lines more pronounced.

357-365 – This analysis is confusing to me and I think needs further explanation. Does the site need to be "high" at all three scales to be in the "high" grouping or are the groupings different based on each spatial scale? Why are the groups the same size, shouldn't the sites determine if they are negative/zero/positive? Are the RSDs below zero in Figure 9d-f due to the station being grouped into the three groups across all scales not for each scale individually? Line 375 starts to answer this but needs to be better included in the body of the paper.

We address this comment in the first section of the document.

393 – Are these completely independent data sources, or does ASO do any QC/shifting of the point cloud based on the SNOTEL site snow depth?

SD is an independent metric. ASO may alter their density values based on in-situ data, but this is reflected in the SWE product, not the SD product.

398 – Removing this bias systematically? Or by each site? I think this would have to occur on a site by site basis since each SNOTEL will act independently from its surroundings.

Yes, we will explain this further. Adjustments would have to be made on a site-by-site basis. This would only be beneficial to sites that have Lidar data. A catch-all adjustment would be more useful since it could work for sites/periods without lidar data, but this method risks deteriorating sites with low RSD vals. We will update the text to clarify this.

435 – Could this be a scale issue? As you mentioned, it is well documented that there is less snow under canopy than open areas depending on the time of year. Is this due to the way RSD is calculated?

We address this comment in the beginning of the document.

465 – another name for the 50 m point measurement, please coordinate these throughout the paper to simplify for the reader.

We will address this.

**Technical Comments**

These technical comments will be addressed.

229 – typo "d" should be a ")"?

247 – missing units on the snow depth range 331

Figure 7b, missing ")"

395 – "location bias."

**References**

Anderson, B. T., J. P. McNamara, H.-P. Marshall, and A. N. Flores (2014), Insights into the physical processes controlling correlations between snow distribution and terrain properties, Water Resour. Res., 50, 4545–4563, doi:10.1002/2013WR013714.

---

## Author Comment (AC2)

**Note: the author responses to comments will be posted in blue. Reviewer comments are in black.**

We would like to extend our sincere gratitude to the reviewer for taking the time to review this manuscript. Their expertise and thoughtful feedback have been invaluable in improving the quality and rigor of our research. Here, we outline the substantive changes we will make to the manuscript based on the comments made by both reviewers. We address line-specific comments in the second section of this document.

**Land cover and topography**

Both reviewers highlighted that our landcover and topography analysis focused on elevation and not other influences on snow distribution such as vegetation and slope / aspect. This was because we only found significant relationships between relative elevation and relative snow depth and not the other factors. We will update the manuscript to include more details about the landcover/topography analysis. We propose the following changes:

- We will add a section to the introduction discussing how the factors that control snow distribution vary with scale. The scales of our analysis are large (km scale) and thus are unable to capture all of the influences on snow distribution.

- We will add the correlation between FVEG/RSD and northness/RSD to the graph in figure 7C. While these relationships are insignificant, this figure will illustrate that we did examine topographic and landcover attributes other than elevation.

- We will expand our discussion to include more details about our expectations for landcover and topography on RSD. The influences of landcover and topography change with scale. Our hypothesis is that at kilometer scales, the effects of elevation are more pronounced than finer scale effects such as vegetation. The aggregation of a physiographic variable to a single value over a kilometer scale may hinder its utility to represent the complex dynamic of a snowpack.

**Organization**

We will condense and re-organize certain parts of the manuscript.

- The literature review presented in the introduction in the two paragraphs from lines 57-80 will be made more succinct. We will summarize the type of research that has been previously conducted in the area but will refrain describing the results in detail. Relevant results will be brought up in the discussion.

- We will remove the equation for RSD and place it within the methods section.

- We will add sub-headings to the analysis section to improve clarity and readability. We will ensure that the order of the analysis section matches the order in which we present the results.

**Temporal consistency analysis**

Both reviewers brought up the qualitative site groupings in Figure 9. We will group the figures quantitatively, with groupings based on the median RSD value. Delineations will be made between sites with mean RSD values of < -0.1m, between -0.1 and 0.1 m, and >0.1 m. An updated figure 9 is presented below:

[Figure]

Second, version 1 of the manuscript did not examine the intra-seasonal variation of relative snow depth. We will add a figure which highlights the intra-seasonal variability of RSD at three selected sites (see below). This figure will be included in the temporal analysis and be included as Figure 10 in the manuscript. This figure 1) highlights that RSD varies seasonally, peaking near peak SWE and reducing in magnitude as melt-out date approaches, and 2) it provides an additional illustration that many sites do have a consistent sign of RSD at all three scales.

[Figure]

Reviewer: Hannah Besso

**General Comments:**

The paper constitutes an important contribution to the field. Snow station data are used for many applications in hydrology. This study adds to our understanding of these stations' representativeness of basin snow quantities and is an important addition to snow hydrology. The scale of the analysis and use of lidar sets it apart from previous studies. However, the authors should explain better and/or reevaluate the temporal analysis. They should also remove the

landcover component from Research Question 3, since the author states in the discussion that the dataset used for this component of the analysis was inadequate. Additionally, the manuscript (especially the Analyses section) should be reorganized or condensed to make the story clearer.

**Organization:** The Introduction includes a deep dive into several relevant papers, whose details are repeated later in the paper. These details should be removed from the Intro. The final paragraphs of the Intro, starting at line 91, would fit better in Methods. The Analysis section seemed to jump from one thing to the next and was confusing to keep track of what you were doing. Either introduce the whole section with a list (could even be bullet points) or make separate section headers with titles that describe what you do for each of these paragraphs/separate analyses. Then in the Results it wasn't always clear which part of the Analysis you were reporting on. It would probably be best to maintain the order in both sections, consistent with the order of the Research Questions.

**Vegetation Impacts:** The lack of a strong vegetation component is a big missing piece of this paper, and it should be highlighted as future work that should be done. I think the paper still stands without a veg component, because the impact of the paper is the bias-correction of the stations, not the reasoning behind that bias. However, understanding the impacts of vegetation on snow quantities at snow stations relative to the surrounding basins is important. Somewhere near the beginning of the manuscript you should acknowledge that vegetation has been proven to impact snow depth, but that your dataset was too coarse to adequately investigate the impacts. Or, if you want to include a vegetation component, you could come up with a simple metric such as distance from station to canopy edge (even just using imagery like on Google Earth). As the manuscript stands currently, the discussion provides good citations of others' work on snow-vegetation interactions, but your Results section reads as if you think there is no impact of vegetation on snow.

**Temporal component of the analysis:** How did you decide on the different groups (that were "typically" low, unbiased, or high)? I'm not fully convinced that these groups are distinct since there's so much overlap in Fig 9D-F.
Pflug and Lundquist (2020) (see Figure 6 of that paper) show that basin snow variability can change throughout a season based on snow covered area and whether it was a 'big' snow year or not. This seems relevant to your Section 3.1, where you argue that a larger range of snow depths increases the maximum magnitude of the RSD. So it would follow that there might be an inter- and intra-season temporal component to changes in RSD at a basin. And Figure 9A-C does show that stations can have a range of RSDs of up to about 50 cm, which seems large relative to your 10cm threshold. I don't find myself convinced that stations are so temporally consistent in their bias that it would be easy to bias-correct them based on just a few lidar flights. I think this would be a huge finding that would have implications for everyone who uses snow station data - I just want to see this proven/investigated a bit more thoroughly. A discussion also might be warranted of whether 3 years of data is enough to develop this relationship.

**Snow depth instead of SWE:** I've been told by people more familiar with CA data than I that snow depth measurements (especially at sites in California, managed by CDEC) are less accurate because they're not maintained or quality controlled as well as the SWE measurements are. I think your reason for using snow depth is valid (given that it's directly measured by the lidar data) but it's worth thinking about and maybe mentioning more than the brief description of your quality

control method.

**Technical Corrections:**
Line 65: Define 'area-mean snow depth' since you use it throughout the paper.

Will ensure to define this and use consistent terminology throughout.

Line 134: "provides no advantage" - higher accuracy of SWE vs snow depth measurements from CA snow stations.

Instead of "provides no advantage," we will say "increases the potential error." We discuss the QC and potential issues of CA snow depth data in line 150.

SWE may have better quality control for the CA snow stations but this does not mean that converting all values to SWE would improve the analysis. Either we would have to use the ASO SWE product (modeled, a black box), or calculate density from the snow station. Calculating density from the snow station requires both SWE and SD, which means we would be still relying on the CA snow depth data as well as adding the uncertainty of assuming a uniform snow density.

Lines 129 - 132: repetitive with the intro. I think you should remove these details from the Intro.

We will delete the justification of our use of snow depth from the introduction.

Lines 150 - 152: Might deserve further discussion.

We will expand on our method of QC. We elected to use CA snow depth data because limiting sites to NRCS stations would reduce the number of basins in CA from 13 to 4. Visual inspection of post-QC station data and comparison with closest lidar SD pixels suggests that the CA data is viable.

Line157: "requires much less storage and computational expense to manage" in comparison, I assume, to the 3 m data set instead of the "range of larger scales" you reference in the previous sentence. Be explicit here.

Will add 'compared to the 3 m datasets' to the end of the sentence.

Lines 154 - 159: How do they produce the 50 m product? Is it derived from the 3 m product?

The products are separate products produced from the Lidar point clouds generated by ASO. The 3 m product is higher resolution than necessary for most operational applications (see lines 187-189).

Lines 161 - 164: DTMs can vary quite a bit in their RMSE, and errors can be spatially variable. For example, areas of a certain DTM with steep slopes or dense vegetation can have larger errors or even systematic bias than in areas that are flat with less vegetation. I think here you could just get away with reporting the error published in Painter et al., 2016. Especially because this is the only lidar dataset you use, so I don't see any added benefit to making generalizations about other lidar products.

We will delete the sentence on lines 162-162 and reduce the lidar error discussion to the citation from Painter et al. (2016).

182 - 183: Confusing sentence, I don't understand what you did.

We will specify that we determine where a point falls within the 'cumulative density function' of the distribution to improve clarity.

Line 184 - 185: Reword this sentence to be more clear. "We compared the snow depth from different data sources at each point location" or something. "Use different data sources to represent the point snow depth value" is confusing.

This sentence will be re-worded.

Line 185 - 187: Can you explicitly define SD? I understand the sentence but had to read it twice to make sure SD was defined.

To make our naming scheme more clear we will change 'snow depth recorded by the snow station' to 'snow station snow depth'

Line 187 - 189: Better suited when you introduce the datasets in 2.1.3.

We will delete these lines.

Paragraph starting with 192: Equation 1 and surrounding text would fit better here than in the introduction.

Line 195: Why did you choose 10 cm? Also, "acceptable" is undefined. Why not say something like "this threshold will change based on the application, but here's why we chose 10cm".

'acceptable' will be changed to 'representative'. The purpose of this section is to discuss that our choice of 10 cm is relatively arbitrary. We will provide further justification for our use of 10 cm as the threshold for representativeness.

Lines 212-213: Confusing sentence and probably unnecessary. This section would likely benefit from a summary sentence at the beginning or end that lists your analyses (see my above comments on organization).

The end of this sentence will be changed to: 'both bilinearly resampled to 50 m to match the resolution of the Lidar data.' We will add subheadings to the analysis section to make the section as a whole flow better.

Line 215: "proportion of representative sites" was confusing, had to read twice. Make it more clear what you're referring to (I assume it refers to 2 paragraphs previous, where you talk about using each cell as the "station" location.)

Noted. We will change 'the proportion of representative sites' to 'the distribution of relative snow depth values.'

Figure 2: I like that you show the different scales using the boxes. But I want the boxes to be different colors than the SD and Elev scales. Also make the SD and Elev gradients different color scales.
For D can you plot the 50m SD and Station SD as vertical lines that intersect your different CDFs

instead of points? Otherwise there are 3 points on this graph representing the same data.

Instead of colored boxes representing the difference scales we will use different line styles (solid, dashed, dotted) as to not interfere with the map colors.

We will plot the point snow depth values on figure 2D as vertical lines instead of individual points.

Line 229: Typo: extra d in "Dd Cumulative"

Will fix.

Figure 3: Same critiques as Fig 2.
Line 233: You use Cumulative Density Function in Figure 2 caption but CDF in Figure 3 caption. Be consistent.

Added an acronym definition for CDF in figure 2.

Line 234: What do you mean by "truncated"? Be explicit.

We meant that the CDF is cut-off by the bounds of the x-axis. Based on the other reviewer's comment we will expand the x-axis to include the entire bounds of the CDF, and remove the sentence about the truncated data.

Line 245: Why include lidar flights that occurred when the study sites were mostly snow-free if this will skew your statistics?

These low-snow flights still provide valuable data on melt out timing and RSD near the melt out date. The proportion of low-snow flights is still only ~20% of the total data points.

Note: you already defined a threshold of 10 cm magnitude RSD. Why so much emphasis on percentiles? This seems like a useful tool in characterizing site variability, but you say it yourself that it's a problematic indicator of representativeness, so emphasize it less. Also, how does the timing of the lidar flights play into this quantile analysis see above comments about the temporal analysis? Do periods of ablation change this relationship?

The percentile we address here is the percentage rank of a point value within the CDF, not the percent difference. We will edit the text to make this more clear.

The purpose of the section is to 1) examine the distribution of snow depths surrounding a snow station, and 2) explain why percentile within the CDF is not an ideal metric for determining representativeness. We will update the first paragraph of section 3.1 to better explain why we conduct this analysis.

We address the comment about intra-seasonal variation in the beginning of the document.

Line 264: stick to cm units for consistency

Will use m instead of cm for consistency with the rest of the text.

Figure 4: Are the vertical lines on your CDFs supposed to represent the 5th and 95th percentile? They don't look like they do (they're all the same width just located in different places - maybe check your code for generating these). If they don't represent those quantiles, I think they need to be labeled/explained.

The last sentence of the caption for figure 4 explains that the vertical black lines represent areas within +/-10 cm of the median snow depth.

Line 289: what are "low sites"? Do you mean "low-biased sites"?

Yes, this will be changed to 'low-biased.'

Lines 291 - 294: I like this summary at the end of the section.
Figure 5: Explain what the gray vertical lines are.

Will add: 'The vertical grey lines at -0.1 m and 0.1 m represent the delineations between low-biased, representative, and high-biased sites.' to the end of the figure caption.

Lines 305 - 308: See my above comments on the vegetation component.

We address this comment in the beginning of the document.

Lines 313-314: perhaps due to vegetation effects.

Certainly. We talk about the potential causes of snow depth bias in the discussion.

Figure 6: The different colors overlap such that they block each other. Is there a way to make both visible via a different type of plot or by using transparency? This is especially a problem at the 0.5 km scale where I think the pink is plotted on top of the blue. Also why is the .5 vertical line lighter than the others? I missed it at first.

We will increase the transparency of the scatter plots in figure 6.

Line 349: "sensing scale"? Does this refer to remote sensing or something else?

Will change to 'spatial coverage' to be consistent with the rest of the manuscript.

Figure 8 caption: use consistent labels. "50 m Lidar pixel" vs. "50 m SD". Also, the 10 cm lines look gray to me instead of black.

We will update the figure caption to be consistent with the naming scheme we use in the rest of the document.

Figure 9: See my above comment about how you grouped the stations. There's a lot of overlap between groups (D-F).

We address this comment at the beginning of the document.

Line 384: I don't think "overrepresent" is the right word here.

We will change 'overrepresent the surrounding area' to 'yield snow depths greater than the areal-mean snow depth'

Lines 386 - 388: this fits better here than in the Intro.

We address this comment at the beginning of the document.

Lines 397 - 398: Rephrase. I think you're saying that any bias correction would need to be site

specific. And do you mean "positively biased" not "oversampling"?

Yes, we will add a second sentence that describes how any bias correction would have to be site specific and require existing lidar data to do so.

We will re-word this sentence to say: "Correcting the bias exhibited by snow station snow depths would mitigate this problem at some sites, but risks deteriorating representativeness at low-biased sites."

Line 400: Instead of a list you should present the infrastructure, flat terrain, etc as components of the location bias. Otherwise this conflicts with the other 2-component list you give in Results.

The list presented in the results is presented to determine if the bias is snow depth is a remnant of sampling error or a true bias in snow depth. The list in line 400 presents possible mechanisms for the bias in snow depth, which we previously determined to not be a result of sampling error. We will add a section to the beginning of this paragraph stating something along the lines of, "Since snow depth bias is not caused by sampling error, what are the potential mechanisms which cause the high bias of 50 m SD values?"

Lines 444 - 445: See above comments on vegetation component of the analysis. Line 465: "pixel", not "point" for the lidar data

We will delete 'closest 50 m Lidar point to represent the snow station SD' and replace it with '50 m SD'

---

## Author Comment (AC3)

Dear Hannah,

Thank you so much for your response. I will respond to your comments here, but I would also be happy to discuss these further offline. I will reach out via email.

**Intra-seasonal RSD variation plot**

Yes, the three sites we selected for the intra-seasonal plots are relatively proximal. We selected these sites because they had the most data spanning different years and periods within the snow season. These sites give the best visualizations of change in relative snow depth throughout the season. Colorado sites generally have fewer data points (max. 2 per year) and are collected at uniform periods in the snow season. We will consider adding sites from Colorado to a second row in the figure. The conclusions would not change based on the addition of these plots. An example of CO sites with the most data points are shown below).

[Figure]

Figure 1. Days to melt vs RSD at three Colorado sites for the 1 km scale. We will consider adding these plots to Figure 10 in the manuscript.

**Vegetation**

Thanks for the clarification on this. It has not been our intention to construe vegetation as unimportant to snow distribution and relative snow depth. Our landcover and vegetation analysis is a secondary aspect of this manuscript and the linear regressions we conduct are quite simple. We wanted to see if any obvious relationships are apparent using widely available landcover and topography datasets. Our plan is to follow this paper up with a more complex analysis of landcover and topography using snow station and lidar data.

The most obvious issue with the relative FVEG metric is that it requires aggregation of all FVEG values within the study area to a single value. This means that any information on forest distribution aside from bulk forest cover is unaccounted for. I think this is a point that we could stress more in the manuscript – the distribution (as opposed to bulk sum) of forest cover is a key variable in the effect of vegetation on snow. We will also stress that the lack of significant relationships between RSD and relative elevation is because the relationship is too complex to be captured by a linear regression of FVEG aggregated to the km scale.

Prior to the next submission of the manuscript we will:
- Analyze other forest cover datasets such as canopy height (from NLCD or ASO).
- Attempt an analysis that uses a metric which includes the variability of vegetation across the landscape.

- Either delete the vegetation cover aspect of question 3 or re-word it to clarify that our analysis focuses on data aggregated to scales that may not be able to cover more complex snow dynamics.

**High bias**
Good point. The benefit of a persistent snowpack at a snow station is mentioned briefly in the introduction (lines 36-38). We will highlight this again in the discussion section.

---

## Author Response (AR1)

**Submission 2, response to comments**

Again, we would like to thank the reviewers and the editor for their thorough review of this work. Here, we outline the primary alterations we made to the manuscript for this second submission. Line-by-line responses follow below.

**Organization**

We have significantly reorganized the manuscript to improve the flow of the document. The majority of changes were made to the methods and results sections.

- The equation for relative snow depth was removed from the Introduction and moved into the methods.

- We have significantly reorganized the methods section.
    - Section 2.1 now focuses on data sources and spatial scales. We have expanded the subheadings of Section 1 to include: Station data, Lidar data, Landcover and topography data, Spatial scales, and Sources of point data.
    - Section 2.2 underwent significant reorganization. We added subheadings and organized the analyses in the order of the research questions they aim to answer. Subheadings include: Snow depth variability, Relative snow depth and representativeness, Landcover and topography analysis, and Consistency of RSD values.

- We reorganized the Results section. Our main goal was to ensure that the order of the results followed the order of the research questions and analyses described in the methods.
    - We split Figure 7 from the original manuscript into two figures. This figure illustrated the results of the expanded scale analysis (0.1-8 km scales) and showed trends of both site representativeness as well as the influence of landcover and topography. The part of Figure 7 that dealt with representativeness was added as its own figure to Section 3.3. The portion of the Figure that dealt with landcover and topography was left in the landcover and topography results section.
    - We moved the point snow depth comparisons to follow Section 3.3 (site representativeness). These results fall under research question 2, so this place is more appropriate in the manuscript. Second, it is a logical next step to show the point snow depth comparisons after demonstrating that different point snow depths yield different results in Section 3.3.

**Land cover and topography**

Both reviewers highlighted that our landcover and topography analysis focused on elevation and not other influences on snow distribution such as vegetation and slope / aspect. This was because we only found significant relationships between relative elevation and relative snow depth and not the other factors. We updated the manuscript as follows:

- We re-phrased question 3 to specify that we examine relative landcover and topography variables to be more specific about the analysis we conduct.

- We have re-worked the first paragraph of section 3.4 to emphasize that southness and fractional vegetation do impact snow depth. To do so, we include the results from regressions of fractional vegetation and southness against snow depth at each site. These results demonstrate that southness and fractional vegetation have significant impacts at snow depth at ~90% of sites. We explain that

the significance breaks down when using the relative values and the reasons for this are further examined in the discussion section.

- We added subheadings to discussion section 4.2, addressing the impacts of elevation, vegetation, and southness on RSD separately. We expanded on the discussion of why we do not observe significant relationships between RSD and relative FVEG / southness.
- We include the figures displaying results from the insignificant landcover and topography variables (FVEG and southness) in the supplement.

**Temporal consistency analysis**

Both reviewers questioned how we decided upon the qualitative site groupings in Figure 9. We now group the figures quantitatively, with groupings based on the median RSD value. Delineations were made between sites with median RSD values of < -0.1m, between -0.1 and 0.1 m, and >0.1 m. The results were not changed when using these delineations as compared to the original method.

Second, we added an analysis of intra-seasonal variation of relative snow depth. To do so, we plotted RSD values against days to snow station melt out for each individual site. We included three example sites as Figure 11 in the manuscript. These sites are all from California. We note that this is required because California sites have the most Lidar surveys as well as surveys which span the greatest range of the snow season, making them more suited for intra-seasonal analysis. Sites from Colorado exhibit similar trends as the examples we provide, but they are not nearly as convincing given that most Colorado sites have only 2 surveys per snow season.

Reviewer: Hannah Besso

**General Comments:**
The paper constitutes an important contribution to the field. Snow station data are used for many applications in hydrology. This study adds to our understanding of these stations' representativeness of basin snow quantities and is an important addition to snow hydrology. The scale of the analysis and use of lidar sets it apart from previous studies. However, the authors should explain better and/or reevaluate the temporal analysis. They should also remove the landcover component from Research Question 3, since the author states in the discussion that the dataset used for this component of the analysis was inadequate. Additionally, the manuscript (especially the Analyses section) should be reorganized or condensed to make the story clearer.

**Organization:** The Introduction includes a deep dive into several relevant papers, whose details are repeated later in the paper. These details should be removed from the Intro. The final paragraphs of the Intro, starting at line 91, would fit better in Methods. The Analysis section seemed to jump from one thing to the next and was confusing to keep track of what you were doing. Either introduce the whole section with a list (could even be bullet points) or make separate section headers with titles that describe what you do for each of these paragraphs/separate analyses. Then in the Results it wasn't always clear which part of the Analysis you were reporting on. It would probably be best to maintain the order in both sections, consistent with the order of the Research Questions.

**Vegetation Impacts:** The lack of a strong vegetation component is a big missing piece of this paper, and

it should be highlighted as future work that should be done. I think the paper still stands without a veg component, because the impact of the paper is the bias-correction of the stations, not the reasoning behind that bias. However, understanding the impacts of vegetation on snow quantities at snow stations relative to the surrounding basins is important. Somewhere near the beginning of the manuscript you should acknowledge that vegetation has been proven to impact snow depth, but that your dataset was too coarse to adequately investigate the impacts. Or, if you want to include a vegetation component, you could come up with a simple metric such as distance from station to canopy edge (even just using imagery like on Google Earth). As the manuscript stands currently, the discussion provides good citations of others' work on snow-vegetation interactions, but your Results section reads as if you think there is no impact of vegetation on snow.

**Temporal component of the analysis:** How did you decide on the different groups (that were "typically" low, unbiased, or high)? I'm not fully convinced that these groups are distinct since there's so much overlap in Fig 9D-F.

Pflug and Lundquist (2020) (see Figure 6 of that paper) show that basin snow variability can change throughout a season based on snow covered area and whether it was a 'big' snow year or not. This seems relevant to your Section 3.1, where you argue that a larger range of snow depths increases the maximum magnitude of the RSD. So it would follow that there might be an inter- and intra-season temporal component to changes in RSD at a basin. And Figure 9A-C does show that stations can have a range of RSDs of up to about 50 cm, which seems large relative to your 10cm threshold. I don't find myself convinced that stations are so temporally consistent in their bias that it would be easy to bias-correct them based on just a few lidar flights. I think this would be a huge finding that would have implications for everyone who uses snow station data - I just want to see this proven/investigated a bit more thoroughly. A discussion also might be warranted of whether 3 years of data is enough to develop this relationship.

**Snow depth instead of SWE:** I've been told by people more familiar with CA data than I that snow depth measurements (especially at sites in California, managed by CDEC) are less accurate because they're not maintained or quality controlled as well as the SWE measurements are. I think your reason for using snow depth is valid (given that it's directly measured by the lidar data) but it's worth thinking about and maybe mentioning more than the brief description of your quality control method.

**Technical Corrections:**
Line 65: Define 'area-mean snow depth' since you use it throughout the paper.

This is now defined and consistent throughout the document.

Line 134: "provides no advantage" - higher accuracy of SWE vs snow depth measurements from CA snow stations.

Instead of "provides no advantage," we say "increases the potential error" (Line 256). We discuss the QC and potential issues of CA snow depth data in the paragraphs below.

SWE may have better quality control for the CA snow stations but this does not mean that converting all values to SWE would improve the analysis. Calculating density from the snow station requires both SWE

and SD, which means we would be still relying on the CA snow depth data as well as adding the uncertainty of assuming a uniform snow density.

Lines 129 - 132: repetitive with the intro. I think you should remove these details from the Intro.

Deleted

Lines 150 - 152: Might deserve further discussion.

The explanation of quality control measures on snow depth data has been expanded.

Line157: "requires much less storage and computational expense to manage" in comparison, I assume, to the 3 m data set instead of the "range of larger scales" you reference in the previous sentence. Be explicit here.

Added 'compared to the 3 m datasets' to the end of the sentence.

Lines 154 - 159: How do they produce the 50 m product? Is it derived from the 3 m product?

The products are separate products produced from the Lidar point clouds generated by ASO.

Lines 161 - 164: DTMs can vary quite a bit in their RMSE, and errors can be spatially variable. For example, areas of a certain DTM with steep slopes or dense vegetation can have larger errors or even systematic bias than in areas that are flat with less vegetation. I think here you could just get away with reporting the error published in Painter et al., 2016. Especially because this is the only lidar dataset you use, so I don't see any added benefit to making generalizations about other lidar products.

We deleted the sentence on lines 162-162 and reduce the lidar error discussion to the citation from Painter et al. (2016).

182 - 183: Confusing sentence, I don't understand what you did.

We now specify that we determine where a point falls within the cumulative density function of the distribution to improve clarity.

Line 184 - 185: Reword this sentence to be more clear. "We compared the snow depth from different data sources at each point location" or something. "Use different data sources to represent the point snow depth value" is confusing.

This sentence will be re-worded.

Line 185 - 187: Can you explicitly define SD? I understand the sentence but had to read it twice to make sure SD was defined.

We changed 'snow depth recorded by the snow station' to 'snow station snow depth'

Line 187 - 189: Better suited when you introduce the datasets in 2.1.3.

Deleted these lines.

Paragraph starting with 192: Equation 1 and surrounding text would fit better here than in the introduction.

RSD equation has been moved to the analyses section.

Line 195: Why did you choose 10 cm? Also, "acceptable" is undefined. Why not say something like "this threshold will change based on the application, but here's why we chose 10cm".

'acceptable' was changed to 'representative'. The purpose of this section is to discuss that our choice of 10 cm is relatively arbitrary. We reference that this is similarly discussed in Meromy et al., 2013. We also add that our inclusion of probability density functions illustrates the distribution of RSD values regardless of what we determine to be representative.

Lines 212-213: Confusing sentence and probably unnecessary. This section would likely benefit from a summary sentence at the beginning or end that lists your analyses (see my above comments on organization).

The analyses section has been completely reorganized.

Line 215: "proportion of representative sites" was confusing, had to read twice. Make it more clear what you're referring to (I assume it refers to 2 paragraphs previous, where you talk about using each cell as the "station" location.)

Noted. We will change 'the proportion of representative sites' to 'the distribution of relative snow depth values.'

Figure 2: I like that you show the different scales using the boxes. But I want the boxes to be different colors than the SD and Elev scales. Also make the SD and Elev gradients different color scales.
For D can you plot the 50m SD and Station SD as vertical lines that intersect your different CDFs instead of points? Otherwise there are 3 points on this graph representing the same data.

- Instead of colored boxes representing the difference scales we use different line styles (solid, dashed, dotted) as to not interfere with the map colors.
- We now plot the point snow depth values on figure 2D as vertical lines instead of individual points.
- **The color of the DEM has been changed to be more distinct from the lidar SD.**

Line 229: Typo: extra d in "Dd Cumulative"

Fixed.

Figure 3: Same critiques as Fig 2.
Line 233: You use Cumulative Density Function in Figure 2 caption but CDF in Figure 3 caption. Be consistent.

Added an acronym definition for CDF in figure 2.

Line 234: What do you mean by "truncated"? Be explicit.

Re-worded to 'cut-off' to be more clear.

Line 245: Why include lidar flights that occurred when the study sites were mostly snow-free if this will skew your statistics?

These low-snow flights still provide valuable data on melt out timing and RSD near the melt out date. The proportion of low-snow flights is still only ~20% of the total data points. As such, we have discussed and elected to keep these flights in the analyses.

Note: you already defined a threshold of 10 cm magnitude RSD. Why so much emphasis on percentiles? This seems like a useful tool in characterizing site variability, but you say it yourself that it's a problematic indicator of representativeness, so emphasize it less. Also, how does the timing of the lidar flights play into this quantile analysis see above comments about the temporal analysis? Do periods of ablation change this relationship?

The percentile we address here is the percent rank of a point value within the CDF, not the percent difference. We make this distinction clearer in the methods. Our results in this section also demonstrate why using the percentile with the CDF is *not* a good metric for representativeness due to the influence of the range of snow depths on the results.

We address the comment about intra-seasonal variation in the beginning of the document.

Line 264: stick to cm units for consistency

The document has been updated to use m instead of cm for consistency with the rest of the text.

Figure 4: Are the vertical lines on your CDFs supposed to represent the 5th and 95th percentile? They don't look like they do (they're all the same width just located in different places - maybe check your code for generating these). If they don't represent those quantiles, I think they need to be labeled/explained.
The last sentence of the caption for figure 4 explains that the vertical black lines represent areas within +/- 10 cm of the median snow depth.

Line 289: what are "low sites"? Do you mean "low-biased sites"?

Yes, changed to 'low-biased.'

Lines 291 - 294: I like this summary at the end of the section.
Figure 5: Explain what the gray vertical lines are.

Added: 'The vertical grey lines at -0.1 m and 0.1 m represent the delineations between low-biased, representative, and high-biased sites.' to the end of the figure caption.

Lines 305 - 308: See my above comments on the vegetation component.

We address this comment in the beginning of the document.

Lines 313-314: perhaps due to vegetation effects.

Certainly. We talk about the potential causes of snow depth bias in the discussion.

Figure 6: The different colors overlap such that they block each other. Is there a way to make both visible via a different type of plot or by using transparency? This is especially a problem at the 0.5 km scale where I think the pink is plotted on top of the blue. Also why is the .5 vertical line lighter than the others? I missed it at first.

We increased the transparency of the scatter plots in figure 6.

Line 349: "sensing scale"? Does this refer to remote sensing or something else?

Changed to 'spatial coverage' to be consistent with the rest of the manuscript.

Figure 8 caption: use consistent labels. "50 m Lidar pixel" vs. "50 m SD". Also, the 10 cm lines look gray to me instead of black.

Updated the figure caption to be consistent with the naming scheme we use in the rest of the document.

Figure 9: See my above comment about how you grouped the stations. There's a lot of overlap between groups (D-F).

We address this comment at the beginning of the document.

Line 384: I don't think "overrepresent" is the right word here.

We will change 'overrepresent the surrounding area' to 'yield snow depths greater than the areal-mean snow depth'

Lines 386 - 388: this fits better here than in the Intro.

We address this comment at the beginning of the document.

Lines 397 - 398: Rephrase. I think you're saying that any bias correction would need to be site specific. And do you mean "positively biased" not "oversampling"?

Yes, we added a second sentence that describes how any bias correction would have to be site specific and require existing lidar data to do so.

We re-worded this sentence to say: "Correcting the bias exhibited by snow station snow depths would mitigate this problem at some sites, but risks deteriorating representativeness at low-biased sites."

Line 400: Instead of a list you should present the infrastructure, flat terrain, etc as components of the location bias. Otherwise this conflicts with the other 2-component list you give in Results.

The list in the results is presented to determine if the bias in snow depth is a remnant of sampling error or a true bias in snow depth. The list in line 400 presents possible mechanisms for the bias in snow depth,

which we previously determined to not be a result of sampling error.
We now state: 'There are multiple possibilities of why station location within a 50 m pixel causes a high-bias' to be more explicit.

Lines 444 - 445: See above comments on vegetation component of the analysis. Line
465: "pixel", not "point" for the lidar data

We deleted 'closest 50 m Lidar point to represent the snow station SD' and replace it with '50 m SD'

**Review Summary**
Herbert et al. use continuous station data and repeat lidar data acquisitions to contextualize the representativeness of station snow depth to the surrounding areas at multiple spatial scales. Through these mixed scale snow depths, the authors additionally work to identify differences in snow depth depending on the sensor and if there are temporal patterns across at each site between each of the lidar acquisition sensors. The primary results indicate that there was no significant difference between the snow depth point measurement and the 3 m lidar snow depth at the station, however significant variability in snow depth between the point locations at the 50 m lidar areal mean were present. Generally, station snow depths area high or representative at 0.5, 1, and 4 km scales while at the 50 m scale snow depth is generally representative with some high. These results indicate that the point station snow depths are representative to the surrounding area, but stations tend to be placed in areas with greater snow depth than their surroundings.

The paper addresses a serious question that has been raised many times on the representativeness of current point measurements of snow across the US at SNOTEL or equivalent sites. These questions have become increasingly important to answer with the proliferation of modelling and remote sensing efforts that utilize the sites as a tuning parameter. The increasing availability of lidar data provides an intriguing opportunity to better define the representativeness of the sites.
While the paper needs refinement prior to publishing, the methods used are appropriate and provide great insight into the stationarity of snow depth point measurements at SNOTEL and CA DWR sites.

I recommend the manuscript for publishing with major modifications, and I provide comments that are necessary to address prior to publishing.

**General Comments**
The manuscript has lots of great details and presents significant scientific findings, however the lack of structure reduces the clarity of the methods and findings. The paper would significantly benefit from additional sub headers within the methods and results sections that guide the reader through the four questions being asked, this would allow the reader to link each of the methods/results to the questions and help guide the reader through the scientific story.
Additionally, while there are lots of great details in the introduction and methods, much of it is repetitive and I suggest the author carefully consider what is included and what distracts from the main points of the paper and should be cut. I recommend the authors condense significant portions of the paper but additional details are needed, specifically in portions of the methods where key data decisions and assumptions are made. The paper would also benefit from consistent use of terminology when referring to snow depth measurements at the varying scales. Finally, there are a few methods that need to be further explained or clarified prior to resubmission.

**Line Review Comments**
33-34 – Tying in the purpose for this paper at this point of the paper is confusing to the reader and would be better suited for the end of the introduction.

Deleted. This paragraph has been adjusted to include more information on the strategic placement of snow stations.

43-89 – It would be beneficial to the article if you revisited these paragraphs and condensed them down. There is a lot of overlap and while the content of each is useful to the reader, it could be condensed to the most relevant information for the paper (use of SNOTEL sites as model validation, assimilation datasets, and extrapolation). Then you could point out the known flaws of using point snow data to represent larger areas and begin to tie in how lidar is a useful tool to fill this knowledge gap in the literature.

We have re-worked this section to increase conciseness to the overarching conclusions made from the relevant existing literature. Though, we believe that this section provides crucial background information to the reader in that it 1) provides context for the uses of snow station data and 2) discusses the results found in previous investigations which assessed representativeness of point snow data. We only include key results that are relevant to this investigation.

91-104 – By simplifying the earlier parts of your introduction, I think you will be better able to set up this study, why it is different and important. Leaving the reader with a clear understanding of what you plan to accomplish in the study and the questions you will be answering.

96 – It might be best to leave the equation for the RSD until the methods, so it is clearer for the reader when specifically describing your methods than having to reference the equation here.

The equation for RSD has been moved to the methods.

103-104 – While I think this is a fine method to take for this research, I do think that this assumption needs to be explained further in the methods.

We added "See section 2.1.2 for further explanation on this decision" at the end of the paragraph to reference where we discuss this decision further.

109-110 – I do not believe this sentence is needed here, it should be mentioned in the introduction that basin wide aerial lidar is becoming more prevalent (ASO, etc.) but it distracts from the study site and data used in the paper when included in this portion of the paper.

Deleted.

123-126 – Do the CA-DWR sites use the same snow depth instrumentation as SNOTEL? Add a citation.

Yes, CA-DWR depth sensors are the same as NRCS sensors based on personal communication. We added a sentence describing that CA-DWR does not provide precision values, but they should be comparable since they use similar hardware.

128-137 – Refer to comment on lines 103-104.

This section provides our justification for using the ASO snow depth product over the SWE product. We will add: "Lidar SWE products use modeled density (Painter et al., 2016), increasing the uncertainty of the SWE product compared to snow depth" to follow the first sentence of the paragraph. This provides an initial justification for our employment of snow depth. The following sentences describe how an assumption of uniform snow density across the landscape provides no advantage over using snow depth as the key variable.

141-149 – What is the positional accuracy of Google Earth imagery? It could be worth working with the NRCS and CA-DWR to ensure you have the correct coordinates for the station and an idea of where the depth sensor is located at each.

We state that Google Earth imagery is available to the fifth decimal place in decimal degrees (~1 m

resolution). We utilized the official coordinates provided by the NRCS and CA-DWR (i.e., the most accurate available data). We found that the location data was not always precisely located on the snow station in Google Earth, necessitating updated coordinates in certain cases. We only used sites that we could visually confirm using satellite imagery.

150-153 – Why does the SNOTEL data not also need further QC? NRCS typically only corrects the daily snow depth data (midnight). What QC methods were taken other than discarding data with a greater than 50 cm difference between hours? Why 50 cm? That seems like a large upper limit that could be tightened to a lesser number that would be more typically seen as a realistic hourly snow accumulation/ablation amount. Would it make sense to use the daily data for this work instead of the hourly data? Assuming that lidar was collected on clear sky days, can we
assume that there was minimal snowfall/snowmelt during the day? Do you use the hour of the flight to compare between the station and the lidar?

The existing text contains a mistake. We *do* employ daily snow depth data, not hourly snow depth data. We switched from hourly to daily data prior to the first submission of this manuscript and did not update the text to reflect the change. We will update the text to specify that we use daily data for NRCS and CA-DWR sites.

Re: QC. We performed QC on all station data. NRCS stations did not require any data deletion, but CA-DWR stations did. The most common error in the data was a rapid shifts of snow depth in one direction followed by a shift in the opposite direction. The figure below illustrates an example of such a shift. We found that accounting for 0.5 m multi-directional shifts in snow depth reduced obvious errors in snow depth data. We updated the text to make this clearer.

[Figure]

174 – Additional sub headers of the different analyses would be beneficial for the reader to understand what methods you are using for each of the separate research questions.

We have reorganized and added subheadings to the analyses section.

186-187 – I think it would be beneficial to the reader if you continued to call these three point measurements by the names listed here throughout the paper.

We ensure consistency with the terminology throughout the paper.

187-188 – This line about 50 m resolution has been included multiple times. I recommend you remove repetitive text throughout the paper unless it is a key result that deserves repetition.

We will delete the justification for using 50 m data in lines 187-188. This sentence is repetitive after the justification we provide in the data section (156-159).

194 – The idea of identifying the representative of the stations to the surrounding area is an important topic, but I wonder if you could use a percentile-based approach that would get the user a percentage of accuracy at each of the scales that they could then judge what "acceptable" is for their use case. This approach would remove the limitations you mention in line 195, although as you mention there are limitations to percentile based analyses.

The limitations that we mention in line 195 are meant to showcase that an acceptable range of error is

dependent on the application for which snow station data is used. This issue would persist for any metric we use to determine representativeness. Meromy et al. (2013) use an acceptable error range of 10% for RSD, but state than any cut-off for acceptability is relatively arbitrary.

In the preparation of this manuscript, we have used absolute magnitude, percent difference, and percentile (rank within the cdf of all snow depths) for classification of RSDs. Meromy et al. were able to use percentile with more success because their analyses did not include surveys near the snow melt-out date (i.e., the range snow depth values was relatively constrained). Our analysis includes data from many different sites, years, and times within the snow season, meaning we have a wide range of snow depth magnitudes (~0-6 m). This range of magnitudes makes a percent difference approach difficult to interpret between data points.

In figure 4 we demonstrate the difficulty of using a percentile-based approach. Areas with uniform snowpack may yield high percentiles for snow depths which are close in magnitude to the point snow depth (e.g., Figure 4B, lines 251-254).

Due to the complications with using percentage and percentile approaches for RSD classification, we elected to use an absolute magnitude approach for RSD classification.

206 – For topography, why are you only analyzing elevation or is aspect also included in the topography analysis? Elevation and landcover are two key portions of a complex relationship between snow depth and mountainous terrain.

We address this comment in the beginning of the document.

212 – This section could be trimmed down to better describe the analysis to the reader succinctly, but additional detail on what you are doing here could also be beneficial. Are you trying to identify the distance at which sites become unrepresentative to the surrounding area? I think this analysis would benefit from some more spatial statistics like variograms (Anderson et al. 2014) or assigning the percentile of representativeness to each scale. Additionally, when stretching to 8km scales, were major topographic features (i.e. ridgelines with low snow due to high winds, topographic basin directions which lead to preferential snow fall, etc.) accounted for in the analysis?

This paragraph has been re-worked with the rest of the analyses section. Figure 6 in the updated document displays how the percentile of representative sites changes with each scale.

221 – Are the other analyses only conducted at one survey date, averages of each date, or are those results also temporal?

We conduct all analyses with all available coincident lidar-snow station data. For each lidar flight in CO and CA between 2021-2023 we find all available snow station data spatially and temporally coincident with the flight and calculate relative snow depth for that date/location. We have updated the text to make this clearer.

Figures 2 and 3 – Although the x-axis scales would be different between figure 2d and 3d, I do think it would be interesting to show the full cdf for both sites.

We have updated the axes in Fig 2d, 3d to extend to 3m so they now capture most values in Spratt Creek (aside from the highest values at the 4 km scale).

245 – Are there surveys available for periods when the stations have snowpack still? Is this the SNOTEL point data of the RSD distributed data? If RSD, at what scale? Can you drop the

surveys with no snow.

Yes, we include all available Lidar data. Some flights occurred in the late snow season, while others were flown closer to peak SD. We do not delete the low snow flights because snow melt-out at the 500 m scale may not reflect snow melt-out at the 4 km scale.

261-264 – How often were the 50 m and SNOTEL measurements ~30 cm different? Keep the units the same (cm or m) when comparing two values.

We updated the document to ensure that all units are reported in meters, not cm. We address the differences between station SD and 50 m SD in later sections (Fig 8).

Figure 4 – It might be helpful for the reader if you added a median line to the plots.

We attempted a median line in Figure 4, but the additional horizontal line makes the figure too busy.

282 – A more thorough comparison of station SD and 50 m lidar SD at the same site could be very informative and help the reader better understand the differences in results between each of the RSDs and the "point" measurement. Jumping right in without this context makes the differences in RMSE hard to understand.

We have moved the point snow depth comparisons up in the document to make the results flow better.

287, 302 – What are the virtual snow stations? What are the Sim. Sites? These all need to be called the same thing if they are, or better explain each in the methods.

The use of 'Sim. Site' in Table 1 was a mistake. We updated to 'virtual site' to be consistent with the rest of the text.

305 – I really appreciate you bringing the question back to the reader, would be helpful if you did this for the other sections as well.

Figure 6 – I think it would be helpful to plot the linear trend lines that you have defined to draw the readers eye to the trends/lack of trends within each. Additionally, are blue points being blocked by pink? It would be interesting to see the difference between the states since we know Colorado and California snowpacks act very differently.

We added transparency to the markers and add trendlines to figure 6.

321 – Why are the sites more likely to have a higher magnitude relative elevation? Is this because sites are typically at low elevation compared to their surroundings?

We added: "due to the increased range of elevations values with scale" to line 322. The reason for increased relative elevation with scale is simply because stations are located in heterogeneous terrain, meaning larger scales are likely to be more different in elevation than smaller scales.

335 – Figure 7c, I am not sure what this is adding, we know that SNOTEL sites are located in mountainous areas with complex terrain features, leading to significant variability (increasing STD) in elevation over larger areas.

The purpose of this is to demonstrate the point in the comment above: larger scales have larger ranges of elevation. These larger elevation ranges lead to increased relative elevation values, in turn decreasing the proportion of representative sites at larer scales. We have split figure 7 into two parts: the parts relating to representativeness and the parts relating to elevation (as described in the beginning of the document).

338 – This section needs to be the first of the results. It sets the stage for all of the other analysis. Then you can identify why you use the point measurement you choose (50 m SD) to complete the remainder of the analysis. Why did you choose to use the 50 m SD for the other analysis?

We have moved this section to just after the representativeness results (Section 3.3) to address the causes for differences in results when using different point snow depths.

Figure 8 – Adding the 1:1 line while making the +-10 cm lines darker would be very helpful for the reader. Again, transparency or making CO and CA their own plots would also be informative.

**We increased the transparency on the scatter dots and made the +/-10 cm lines more pronounced.**

357-365 – This analysis is confusing to me and I think needs further explanation. Does the site need to be "high" at all three scales to be in the "high" grouping or are the groupings different based on each spatial scale? Why are the groups the same size, shouldn't the sites determine if
they are negative/zero/positive? Are the RSDs below zero in Figure 9d-f due to the station being grouped into the three groups across all scales not for each scale individually? Line 375 starts to answer this but needs to be better included in the body of the paper.

We address this comment in the first section of the document.

393 – Are these completely independent data sources, or does ASO do any QC/shifting of the point cloud based on the SNOTEL site snow depth?

SD is an independent metric. ASO may alter their density values based on in-situ data, but this is reflected in the SWE product, not the SD product.

398 – Removing this bias systematically? Or by each site? I think this would have to occur on a site by site basis since each SNOTEL will act independently from its surroundings.

Yes, we will explain this further. Adjustments would have to be made on a site-by-site basis. This would only be beneficial to sites that have Lidar data. A catch-all adjustment would be more useful since it could work for sites/periods without lidar data, but this method risks deteriorating sites with low RSD vals. We will update the text to clarify this.

435 – Could this be a scale issue? As you mentioned, it is well documented that there is less
snow under canopy than open areas depending on the time of year. Is this due to the way RSD is calculated?

We address this comment in the beginning of the document.

465 – another name for the 50 m point measurement, please coordinate these throughout the paper to simplify for the reader.

We ensure consistency of terminology throughout the document.

**Technical Comments**

These technical comments have been addressed.

229 – typo "d" should be a ")"?

247 – missing units on the snow depth range 331

Figure 7b, missing ")"

395 – "location bias."

---

## Referee Report (RR1)

Herbert et al. – Review #2

Wyatt Reis

I appreciate the effort the authors have put into addressing my first round of comments. I feel that they have adequately addressed those comments and the changes have improved the manuscript. Here I have identified a few minor comments that should be addressed prior to publishing.

Section 2.1.4 – Would it make more sense for this section to fall under section 2.2 since this section describes an analysis method not the source of data?

Figure 3 – I recommend writing out the full caption of the figure instead of relying on the Figure 2 caption. This will cause the figure captions to be repetitive, but it would be a more thorough solution. Another solution could be to combine these figures and make them an eight-panel plot. This would reduce the amount of text while still allowing the key takeaways from the figures to remain.

 Section 2.2.4 – Would it make more sense for this section to be a sub section of Section 2.2.2? Jumping from RSD in Section 2.2.2 to Landcover and topography (Section 2.2.3), then back to RSD (Section 2.2.4) is jarring for the reader. The same is true for the results section.

Line 266 – Is "Lidar" referring to the 50 m SD?

Line 270-273 – Here and elsewhere (Line 371-374 is another example) are sections of text in the results that read more like interpretation and would be better suited for the discussion. I recommend the authors move portions of interpretive text from the results to the discussion.

Figure 4 – This caption could be revised to improve the readability. Could rewrite to "Histogram plot of the 5-95th percentile Lidar snow depth values around snow stations in (a) Colorado (138 sites) and (f) California (338 sites). Cumulative density function plots at select sites in (b-e) Colorado and (g-j) California spanning low to high snow depth variability...."

Line 291 – Could remove "(described in Section 2.2.2)." With the improved structure of the paper, you have implemented the reader can easily find the appropriate methods section.

Section 3.3 – Most other section headers only capitalize the first word. Make consistent.

Section 3.5 – Update all figure references to Figure 10.

---

## Referee Report (RR2)

Hannah Besso
bessoh2@uw.edu

The authors have responded thoroughly to my review, improving the manuscript considerably. I don't need to see the manuscript again. It contributes significantly to our understanding of Snotel stations and snow variability in the Western U.S. I appreciate the extra analyses the authors have added since I last read the paper and I think the snow science community will benefit from this work. I have a list of suggested edits below, but none are required for publication. The one point I would like to emphasize for the authors to address is the framing around the motivation for using the 50 m Lidar pixel as a 'point' measurement. My opinion is that this value should be referred to as a 'pixel' instead of as a 'point', and that the motivation for this part of the analysis should be more clear.

Line 11: Sentence starting with "The nearest 50 m Lidar pixels had lower bias…" explicitly state what this bias is calculated in comparison to. This sentence makes it sound like you think 50 m lidar pixels are more accurate than snotel stations, so clarify what exactly you mean. A 50 m lidar pixel should be more representative of a larger area just by the nature of the 50 m pixel size as opposed to a point (or 3x3, as the case may be) measurement.

Line 14: Might want to mention aspect and vegetation here as the other variables you looked at.

Line 20: Maybe say 'comprises about half' instead of the 'majority'. The percentage is 53% according to Li et al., 2017, and the margin of error is likely such that there's a few percentage points of error on either side of that.

Line 86: Erroneous question mark after label for research question 4.

Section 2.1.2: I think you need to mention the relationship between the 3m and 50m ASO products. Is the 50m derived from the 3m gridded product, or are these independently calculating using the original lidar point cloud?

Line 178: It's unclear here how this second analysis of up to 8 km differs from the analysis explained in the previous paragraph up to 4 km.

Line 185: I think this needs slightly more clarification. Be clear about what the purpose of the 50 m Lidar pixel is, since the obvious comparison would just be the snotel point to the 3m Lidar pixel. So why do you use the 50 m Lidar grid cell throughout this paper? How does this help inform Snotel users? See my comment about this above, and my comment on the Conclusions.

Line 186: It seems a little odd to me to refer to point data at different scales. Point data is usually just a single point, not an aggregate of a 50m area. Is there a scenario where modelers would use a larger gridcell as a point measurement? If so, include something about this in your motivation for this section. This was an issue for me throughout the paper - if you feel you have

a strong reason for this then just be clear about it. Otherwise I don't think there's any downside to referring to this as a pixel instead of as a point.

Figures 2 & 3: For subfig A, I would recommend a non-diverging color scheme, like white to blue (using colors associated with snow), instead of one that emphasizes 0 values as very dark red.

Line 260: Is the 0.3 - 0.6 m depths true for all flights or just those near peak depth?

Lines 267 - 270: I'm not sure what these extra examples add. I don't think you need them. If you do keep them, I would include them in your reference on Line 265 where you reference Fig 4b,c and be more clear about what insight they add (are you saying they are additional examples of sites with a small range of values, or do they add something by being examples of sites with 'medium' ranges?).

Lines 264 - 277: I think a more clear way of illustrating this idea would be to combine these two paragraphs. I'm thinking along the lines of:

"The CDF plots demonstrate a range of possible scenarios created from different snow depth distributions. Sites characterized by lower snow depth variability (Fig. 4b, c, g, h) are less likely to have point snow depths far from the median due to the limited range of snow depths, while sites with higher snow depth variability (Fig. 4d, e, i, j) allow for greater differences between the median and point snow depth. For example, at the Michigan Creek Snotel site (Fig. 4b) the Lidar and Snotel point values correspond to the 7th and 95th percentiles, yet both values are within 0.1 m depth of the median value. Conversely, at Scotch Creek (Fig. 4e) the station SD (blue marker) is not representative as it is 0.46 m greater than the median snow depth value, near the 100th percentile of the areal distribution."

Line 294: I would change for clarity to: "Root-mean-square error (RMSE) is 0.46, 0.48, and 0.54 m for the same respective spatial scales."

Fig. 5: Looks like you lumped together results for CA and CO here. Do these distributions look similar between the two locations? In the other figures you separate the two, so maybe mention in the caption that results are for all sites.

Line 307: "when using 50 m SD as the point value" - noting the use of 'point' instead of 'pixel' in case you decide to change this terminology.

Line 324: Missing the word 'at'. Should read: "are that [at] the smaller scales".

Figure 6: Would prefer a more descriptive y axis title.

Line 346: Instead of 'higher snow depth', which is a weird phrase, I recommend: 'Thus, we conclude that the high bias reported by the station SD and 3 m SD is a result of true differences in snow depth at the station locations compared to the surrounding 50 m area.'

Section 3.4: I don't think you mention in the text that you use the 50 m Lidar pixel instead of the station, although this is in the caption for figure 8. Worth mentioning in the first paragraph.

Line 398: Do you mean higher magnitude relative elevation differences?

Line 408: When you refer here to the 'previous section', does this refer to the previous paragraph about timing or the rest of the results section? If it's referring to the previous paragraph, change the wording to 'the above paragraphs' since they're within the same section. If you're referring to the other results in the results section, make 'previous section' plural.

Line 424: Rewrite this sentence - it's ~50% of stations that exceed areal-mean by more than 10cm, and the percentage of stations that are not spatially representative are more than this. So be more precise with this wording perhaps by removing 'were not spatially representative'.

Line 425: Likewise, be more precise here: It's the finding that snow stations tend to be high-biased that is not unprecedented, not the tendency itself.

Line 435: Used 'point' to refer to the 50 m lidar pixel again here.

Section 41: Consider using another word instead of 'elevated', which makes me think it's related to elevation, and the term 'elevated snow depth' doesn't really make sense. Maybe instead say, for example on line 445: 'suggested that deeper snow at stations compared to the surrounding area was a result of flat terrain…'

Line 447-448: Fix the usage of commas.

Line 457: First sentence of 4.2: could use a few more citations for this broad statement.

Line 492: You mention the need for high resolution - could be helpful to restate the resolution of the vegetation product you used so we can compare that to the <5m resolution needed.

Line 493: Edit this sentence for clarity. Perhaps something like 'Here we used relative fractional vegetation as the metric to describe vegetation dynamics, which reduces these dynamics to a single value. This value may be insufficient to…'

Line 505: To be more succinct, use 'it is possible' instead of 'it is a possibility'.

Line 510: Use a hyphen: 'within-season'. Or just say 'inter- and intra-annual consistency'.

Line 530: Be more precise. The term 'When using the 50 m SD' leaves ambiguity. State what you are comparing more clearly.

Lines 530 - 538: I think what you're saying is that the 50 m Lidar pixel is a better representation of large-scale snow depth than the snotel data is, so if we can bias-correct snotel to better match the 50 m lidar pixel, then the bias-corrected snotel data will therefore better represent large-scale depth as well. I didn't feel like this motivation was clear throughout the paper (if I am indeed interpreting this correctly). I was unsure throughout the paper as to why you were using a 50 m Lidar pixel as 'point' data in addition to the snotel data. I think the framing around this needs to be better: you're proving what we intuitively know - which is that when you incorporate data from a larger area of the basin (50 m lidar pixel instead of 3m-scale snotel point), you will get a better estimate of the depth throughout the basin. Then you show that there is a temporally-consistent offset at the snotel point compared to that 50 m pixel - so that we can bias-correct the snotel data to better represent snow depth within a larger area. Throughout the paper, I felt like you were building a case for people to use 50 m lidar pixels instead of snotel data, which of course we can't do in places/times without lidar. Regardless of whether my interpretation of this is correct, please add clarity throughout the paper about the motivations here. Section 2.1.5 would be a good place to add detail.

Lines 551 - 554: This might be journal-specific, but it would make more sense to me if this was moved to the Data Availability section.

---

## Author Response (AR2)

We would like to thank the reviewers for this second evaluation of our manuscript. These insights and suggestions have been instrumental in further refining our work, and we are grateful for the constructive guidance. We address the comments and the changes to the manuscript we have made below.

Second Review for Herbert et al.
Hannah Besso
bessoh2@uw.edu

The authors have responded thoroughly to my review, improving the manuscript considerably. I don't need to see the manuscript again. It contributes significantly to our understanding of Snotel stations and snow variability in the Western U.S. I appreciate the extra analyses the authors have added since I last read the paper and I think the snow science community will benefit from this work. I have a list of suggested edits below, but none are required for publication. The one point I would like to emphasize for the authors to address is the framing around the motivation for using the 50 m Lidar pixel as a 'point' measurement. My opinion is that this value should be referred to as a 'pixel' instead of as a 'point', and that the motivation for this part of the analysis should be more clear.

Line 11: Sentence starting with "The nearest 50 m Lidar pixels had lower bias…" explicitly state what this bias is calculated in comparison to. This sentence makes it sound like you think 50 m lidar pixels are more accurate than snotel stations, so clarify what exactly you mean. A 50 m lidar pixel should be more representative of a larger area just by the nature of the 50 m pixel size as opposed to a point (or 3x3, as the case may be) measurement.

Changed to: 'more often representative of the areal-mean snow depth than coincident stations'

Line 14: Might want to mention aspect and vegetation here as the other variables you looked at.

Added a sentence: 'Relative values of vegetation and southness did not have significant impacts on site representativeness.'

Line 20: Maybe say 'comprises about half' instead of the 'majority'. The percentage is 53% according to Li et al., 2017, and the margin of error is likely such that there's a few percentage points of error on either side of that.

Good point – changed.

Line 86: Erroneous question mark after label for research question 4.

Deleted.

Section 2.1.2: I think you need to mention the relationship between the 3m and 50m ASO products. Is the 50m derived from the 3m gridded product, or are these independently calculating using the original lidar point cloud?

Added a sentence describing that both datasets are produced by taking the difference between snow-on and snow-off points clouds.

Line 178: It's unclear here how this second analysis of up to 8 km differs from the analysis explained in the previous paragraph up to 4 km.

For this analysis we wanted a higher resolution dataset of the relationship between representativeness and scale (e.g. Fig. 6) as well as to determine if the trends we observe continue beyond the 4 km scale. We added a sentence in this paragraph to more clearly explain our intent.

Line 185: I think this needs slightly more clarification. Be clear about what the purpose of the 50 m Lidar pixel is, since the obvious comparison would just be the snotel point to the 3m Lidar pixel. So why do you use the 50 m Lidar grid cell throughout this paper? How does this help inform Snotel users? See my comment about this above, and my comment on the Conclusions.

We have added a sentence to the end of the paragraph describing why we primarily use 50 m SD and station SD. Station SD provides a true comparison of station data to the surrounding area while 50 m SD assesses the more general location of the snow station within the landscape. One of our key results is that snow station data is high-biased, which is reduced when using the 50 m SD. This suggests that the placement of snow stations within the landscape is only slightly high-biased (Fig. 6). We would not be able to conclude this result if we did not include 50 m SD in our analysis.

Line 186: It seems a little odd to me to refer to point data at different scales. Point data is usually just a single point, not an aggregate of a 50m area. Is there a scenario where modelers would use a larger gridcell as a point measurement? If so, include something about this in your motivation for this section. This was an issue for me throughout the paper - if you feel you have a strong reason for this then just be clear about it. Otherwise I don't think there's any downside to referring to this as a pixel instead of as a point.

As described above, we have altered the wording in Section 2.1.5 and no longer use the term point data. The section title is now 'Data representing the snow station'. We explain our reasoning for the use of 50 m SD in our analyses in the comment above.

Figures 2 & 3: For subfig A, I would recommend a non-diverging color scheme, like white to blue (using colors associated with snow), instead of one that emphasizes 0 values as very dark red.

Changed to white-blue color scheme for figures 2 and 3.

Line 260: Is the 0.3 - 0.6 m depths true for all flights or just those near peak depth?

The split of lidar flights of lidar flights in CO is generally near peak SWE or late ablation season. Based on this, the 0.3-0.6 mode primarily consists of near peak SWE lidar flights.

Lines 267 - 270: I'm not sure what these extra examples add. I don't think you need them. If you do keep them, I would include them in your reference on Line 265 where you reference Fig 4b,c and be more clear about what insight they add (are you saying they are additional examples of sites with a small range of values, or do they add something by being examples of sites with 'medium' ranges?).

Response below.

Lines 264 - 277: I think a more clear way of illustrating this idea would be to combine these two paragraphs. I'm thinking along the lines of:

"The CDF plots demonstrate a range of possible scenarios created from different snow depth distributions. Sites characterized by lower snow depth variability (Fig. 4b, c, g, h) are less likely to have point snow depths far from the median due to the limited range of snow depths, while sites with higher snow depth variability (Fig. 4d, e, i, j) allow for greater differences between the median and point snow depth. For example, at the Michigan Creek Snotel site (Fig. 4b) the Lidar and Snotel point values correspond to the 7th and 95th percentiles, yet both values are within 0.1 m depth of the median value. Conversely, at Scotch Creek (Fig. 4e) the station SD (blue marker) is not representative as it is 0.46 m greater than the median snow depth value, near the 100th percentile of the areal distribution."

We have updated this section according to the suggestions. The two paragraphs have been condensed into one for increased conciseness.

Line 294: I would change for clarity to: "Root-mean-square error (RMSE) is 0.46, 0.48, and 0.54 m for the same respective spatial scales."

Updated.

Fig. 5: Looks like you lumped together results for CA and CO here. Do these distributions look similar between the two locations? In the other figures you separate the two, so maybe mention in the caption that results are for all sites.

Line 307: "when using 50 m SD as the point value" - noting the use of 'point' instead of 'pixel' in case you decide to change this terminology.

We have removed 'point' and now say 'when using 50 m SD to represent that station.'

Line 324: Missing the word 'at'. Should read: "are that [at] the smaller scales".

Fixed.

Figure 6: Would prefer a more descriptive y axis title.

Changed y-axis to 'percent of sites' to be more descriptive.

Line 346: Instead of 'higher snow depth', which is a weird phrase, I recommend: 'Thus, we conclude that the high bias reported by the station SD and 3 m SD is a result of true differences in snow depth at the station locations compared to the surrounding 50 m area.'

We updated the sentence according to the suggestion.

Section 3.4: I don't think you mention in the text that you use the 50 m Lidar pixel instead of the station, although this is in the caption for figure 8. Worth mentioning in the first paragraph.

We added that RSD was calculated using 50 m SD in the second sentence of the first paragraph.

Line 398: Do you mean higher magnitude relative elevation differences?

Yes, updated.

Line 408: When you refer here to the 'previous section', does this refer to the previous paragraph about timing or the rest of the results section? If it's referring to the previous paragraph, change the wording to 'the above paragraphs' since they're within the same section. If you're referring to the other results in the results section, make 'previous section' plural.

Changed to 'the above paragraphs'

Line 424: Rewrite this sentence - it's ~50% of stations that exceed areal-mean by more than 10cm, and the percentage of stations that are not spatially representative are more than this. So be more precise with this wording perhaps by removing 'were not spatially representative'.

We updated the sentence to: 'We found that station SDs exceeded the areal-mean snow depth by at least 10 cm in ~50% of cases at all scales.'

Line 425: Likewise, be more precise here: It's the finding that snow stations tend to be high-biased that is not unprecedented, not the tendency itself.

Changed to 'finding'

Line 435: Used 'point' to refer to the 50 m lidar pixel again here.

Re-worded to remove 'point' from the sentence.

Section 41: Consider using another word instead of 'elevated', which makes me think it's related to elevation, and the term 'elevated snow depth' doesn't really make sense. Maybe instead say, for example on line 445: 'suggested that deeper snow at stations compared to the surrounding area was a result of flat terrain…'

Changed 'elevated snow depths' to 'deeper snow' in two instances in Section 4.1.

Line 447-448: Fix the usage of commas.

The sentence has been re-worded.

Line 457: First sentence of 4.2: could use a few more citations for this broad statement.

Added three more citations.

Line 492: You mention the need for high resolution - could be helpful to restate the resolution of the vegetation product you used so we can compare that to the <5m resolution needed.

We added that our 30 m FVEG dataset would not capture these dynamics.

Line 493: Edit this sentence for clarity. Perhaps something like 'Here we used relative fractional vegetation as the metric to describe vegetation dynamics, which reduces these dynamics to a single value. This value may be insufficient to…'

Updated according to suggestions.

Line 505: To be more succinct, use 'it is possible' instead of 'it is a possibility'.

Updated

Line 510: Use a hyphen: 'within-season'. Or just say 'inter- and intra-annual consistency'.

Updated

Line 530: Be more precise. The term 'When using the 50 m SD' leaves ambiguity. State what you are comparing more clearly.

Changed to: 'When relative snow depth is calculated with 50 m SD'

Lines 530 - 538: I think what you're saying is that the 50 m Lidar pixel is a better representation of large-scale snow depth than the snotel data is, so if we can bias-correct snotel to better match the 50 m lidar pixel, then the bias-corrected snotel data will therefore better represent large-scale depth as well. I didn't feel like this motivation was clear throughout the paper (if I am indeed interpreting this correctly). I was unsure throughout the paper as to why you were using a 50 m Lidar pixel as 'point' data in addition to the snotel data. I think the framing around this needs to be better: you're proving what we intuitively know - which is that when you incorporate data from a larger area of the basin (50 m lidar pixel instead of 3m-scale snotel point), you will get a better estimate of the depth throughout the basin. Then you show that there is a temporally-consistent offset at the snotel point compared to that 50 m pixel - so that we can bias-correct the snotel data to better represent snow depth within a larger area. Throughout the paper, I felt like you were building a case for people to use 50 m lidar pixels instead of snotel data, which of course we can't do in places/times without lidar. Regardless of whether my interpretation of this is correct, please add clarity throughout the paper about the motivations here. Section 2.1.5 would be a good place to add detail.

The main point here is that there is a systematic high bias at snow stations. We observe a high bias of the snow station compared to the surrounding area at all scales (50 m – 8 km). This is a finding that is important to consider when using snow station data for validation at any scale.

We added a sentence to the conclusion that the high bias at snow stations should be considered when using station data for validation.

We also added a sentence to section 2.1.4 (formerly 2.1.5) describing that we primarily use 50 m SD and station SD because assessing the two spatial scales provides results about the station data itself (station SD) as well as the more general location of the station (50 m SD).

Lines 551 - 554: This might be journal-specific, but it would make more sense to me if this was

moved to the Data Availability section.

We moved information about the data sources to the Data Availability section.

Herbert et al. – Review #2
Wyatt Reis

I appreciate the effort the authors have put into addressing my first round of comments. I feel that they have adequately addressed those comments and the changes have improved the manuscript. Here I have identified a few minor comments that should be addressed prior to publishing.

Section 2.1.4 – Would it make more sense for this section to fall under section 2.2 since this section describes an analysis method not the source of data?

We added the spatial scale description to the beginning of Section 2.2. First, we describe the spatial scales of our analyses, then in the subheadings 2.2.X, we describe the specific analyses we conduct to answer the research questions.

Figure 3 – I recommend writing out the full caption of the figure instead of relying on the Figure 2 caption. This will cause the figure captions to be repetitive, but it would be a more thorough solution. Another solution could be to combine these figures and make them an eight-panel plot. This would reduce the amount of text while still allowing the key takeaways from the figures to remain.

We now write out the complete caption for both figures.

Section 2.2.4 – Would it make more sense for this section to be a sub section of Section 2.2.2? Jumping from RSD in Section 2.2.2 to Landcover and topography (Section 2.2.3), then back to RSD (Section 2.2.4) is jarring for the reader. The same is true for the results section.

We have re-ordered the research questions so that question 3 now addresses temporal consistency and question 4 addresses the influence of topography. We reordered the analyses, results, and discussion in accordance with the new order.

Line 266 – Is "Lidar" referring to the 50 m SD?

Yes, we updated the sentence to '50 m SD and station SD values'

**Line 270-273 – Here and elsewhere (Line 371-374 is another example) are sections of text in the results that read more like interpretation and would be better suited for the discussion. I recommend the authors move portions of interpretive text from the results to the discussion.**

We have moved lines 371-374 to the discussion section.

We believe it is most effective to leave lines 270-273 where they are. In these lines we demonstrate why cdf percentile is not a good metric to measure site representativeness. Thus, when we assess site representativeness in the following section, the reader knows why we have discarded the percentile metric.

Figure 4 – This caption could be revised to improve the readability. Could rewrite to "Histogram plot of the 5-95th percentile Lidar snow depth values around snow stations in (a) Colorado (138 sites) and (f) California (338 sites). Cumulative density function plots at select sites in (b-e) Colorado and (g-j) California spanning low to high snow depth variability   "

Caption has been updated according to the suggestions.

Line 291 – Could remove "(described in Section 2.2.2)." With the improved structure of the paper, you have implemented the reader can easily find the appropriate methods section.

Removed.

Section 3.3 – Most other section headers only capitalize the first word. Make consistent.

Changed Section 3.3 to lower case after the first word to be consistent with the rest of the document.

Section 3.5 – Update all figure references to Figure 10.

Updated.